EMBO
Molecular Medicine

# ADAM17 selectively activates the IL-6 trans-signaling/ERK MAPK axis in KRAS-addicted lung cancer

Mohamed I Saad[1,2] [ID], Sultan Alhayyani[1,2], Louise McLeod[1,2], Liang Yu[1,2], Mohammad Alanazi[1,2], Virginie Deswaerte[1,2], Ke Tang[1,2], Thierry Jarde[3,4,5], Julian A Smith[6,7], Zdenka Prodanovic[8], Michelle D Tate[1,2], Jesse J Balic[1,2], D Neil Watkins[9], Jason E Cain[2,5], Steven Bozinovski[10], Elizabeth Algar[5,11], Tomohiro Kohmoto[12,13], Hiromichi Ebi[14,15], Walter Ferlin[16], Christoph Garbers[17], Saleela Ruwanpura[1,2], Irit Sagi[18], Stefan Rose-John[19] & Brendan J Jenkins[1,2,* [ID]

## Abstract

Oncogenic *KRAS* mutations are major drivers of lung adenocarcinoma (LAC), yet the direct therapeutic targeting of KRAS has been problematic. Here, we reveal an obligate requirement by oncogenic KRAS for the ADAM17 protease in LAC. In genetically engineered and xenograft (human cell line and patient-derived) *Kras*[G12D]-driven LAC models, the specific blockade of ADAM17, including with a non-toxic prodomain inhibitor, suppressed tumor burden by reducing cellular proliferation. The pro-tumorigenic activity of ADAM17 was dependent upon its threonine phosphorylation by p38 MAPK, along with the preferential shedding of the ADAM17 substrate, IL-6R, to release soluble IL-6R that drives IL-6 trans-signaling via the ERK1/2 MAPK pathway. The requirement for ADAM17 in *Kras*[G12D]-driven LAC was independent of bone marrow-derived immune cells. Furthermore, in *KRAS* mutant human LAC, there was a significant positive correlation between augmented phospho-ADAM17 levels, observed primarily in epithelial rather than immune cells, and activation of ERK and p38 MAPK pathways. Collectively, these findings identify ADAM17 as a druggable target for oncogenic *KRAS*-driven LAC and provide the rationale to employ ADAM17-based therapeutic strategies for targeting *KRAS* mutant cancers.

**Keywords** ADAM17; ERK MAPK; IL-6 trans-signaling; KRAS; lung adenocarcinoma
**Subject Categories** Cancer; Respiratory System

## Introduction

Non-small-cell lung cancer (NSCLC) accounts for ~85% of all cases of lung cancer, the most lethal cancer worldwide (Wong *et al*, 2017). The majority of NSCLC patients present with lung adenocarcinoma (LAC) at an advanced stage, with indiscriminate treatment options restricted to surgery, chemotherapy, and/or radiation therapy which are associated with a high risk of tumor re-occurrence and poor overall 5-year relative survival rates (15–20%) (NSCLC Meta-analysis Collaborative Group, 2014; Wong *et al*, 2017). To combat these

1 Centre for Innate Immunity and Infectious Diseases, Hudson Institute of Medical Research, Clayton, Vic., Australia
2 Department of Molecular and Translational Sciences, Faculty of Medicine, Nursing and Health Sciences, Monash University, Clayton, Vic., Australia
3 Cancer Program, Monash Biomedicine Discovery Institute, Clayton, Vic., Australia
4 Department of Anatomy and Developmental Biology, Monash University, Clayton, Vic., Australia
5 Centre for Cancer Research, Hudson Institute of Medical Research, Clayton, Vic., Australia
6 Department of Surgery, School of Clinical Sciences at Monash Health, Monash University, Clayton, Vic., Australia
7 Department of Cardiothoracic Surgery, Monash Health, Clayton, Vic., Australia
8 Monash Biobank, Monash Health, Clayton, Vic., Australia
9 The Kinghorn Cancer Centre, Garvan Institute of Medical Research, Darlinghurst, NSW, Australia
10 School of Health and Biomedical Sciences, RMIT University, Bundoora, Vic., Australia
11 Genetics and Molecular Pathology Laboratory, Monash Health, Clayton, Vic., Australia
12 Department of Human Genetics, Tokushima University Graduate School of Medicine, Tokushima, Japan
13 Division of Molecular Genetics, Aichi Cancer Center Research Institute, Nagoya, Japan
14 Division of Molecular Therapeutics, Aichi Cancer Center Research Institute, Nagoya, Japan
15 Division of Advanced Cancer Therapeutics, Nagoya University Graduate School of Medicine, Nagoya, Japan
16 NovImmune SA, Geneva, Switzerland
17 Department of Pathology, Medical Faculty, Otto-von-Guericke University, Magdeburg, Germany
18 Department of Biological Regulation, Weizmann Institute of Science, Rehovot, Israel
19 Institute of Biochemistry, Christian-Albrechts-University, Kiel, Germany
*Corresponding author. Tel: +61 3 8572 2740; E-mail: brendan.jenkins@hudson.org.au

issues, the last decade has seen the advent of alternate treatment modalities, such as tyrosine kinase inhibitors (TKIs) as a first-line targeted therapy option in NSCLC patients positive for epidermal growth factor receptor (EGFR) constitutive activating mutations (Hirsch et al, 2017). However, the clinical response to such TKIs is invariably short-lived due to acquired drug resistance (Hirsch et al, 2017). More recently, combined immunotherapy with PD-1/PD-L1 plus CTLA-4 checkpoint inhibitors has been an area of intense clinical activity in NSCLC, although the clinical benefits of such immune system modulation remain unclear (Hirsch et al, 2017).

One of the most extensively studied oncogenes in human LAC is KRAS. Activating KRAS mutations are found in one-third of LAC patients, with a high frequency in codon 12 (e.g., G12D) that are linked to tobacco smoke exposure and poor survival prognosis (Ahrendt et al, 2001; Román et al, 2018). A definitive role for KRAS in the molecular pathogenesis of LAC is evidenced by the spontaneous development of LAC in mouse models genetically engineered to conditionally express the oncogenic $Kras^{G12D}$ allele in the airway epithelium (DuPage et al, 2009). Despite these observations, the development and clinical implementation of specific KRAS inhibitors have proven technically challenging, and there is now a pressing need to identify druggable cooperating partners of oncogenic KRAS (Cox et al, 2014).

The a disintegrin and metalloproteinase (ADAM) family member, ADAM17, is a type I transmembrane protease that drives the limited proteolysis of over 80 cell membrane-bound cytokines, chemokines, growth factors, adhesion molecules, and their receptors (Zunke & Rose-John, 2017). Following its initial production as an inactive pro-form, the functional maturation and activation of ADAM17 are regulated by a complex series of post-translational processing and trafficking events. These include furin-mediated removal of the inhibitory prodomain, interactions with inactive members of the rhomboid family, protein disulfide isomerase and phosphatidylserine, and phosphorylation of its cytoplasmic domain by serine/threonine kinases (e.g., PKCα/δ, ERK and p38 MAPKs, and PLK2) (Zunke & Rose-John, 2017). Among ADAM17 substrates, IL-6R, TNFα, Notch1, and EGFR family ligands (e.g., TGFα, amphiregulin, epiregulin, and neuregulin-1) promote the proliferation, survival, migration, and/or invasion of tumor cells (Zunke & Rose-John, 2017), thus suggesting a prominent role for ADAM17 in cancer. In support of this notion, oncogenic KRAS can modulate ADAM17-induced shedding of EGFR family ligands during colorectal and pancreatic cancers, and genetic targeting of ADAM17 in mouse models for these cancers suppresses tumorigenesis by abrogating EGFR family ligand production (Van Schaeybroeck et al, 2011; Ardito et al, 2012; Schmidt et al, 2018). Small molecule inhibitors (SMIs) of ADAM17 also suppress the growth of various human cancer cell line xenografts (Zhou et al, 2006; Fridman et al, 2007). However, the clinical efficacy of SMIs has been hampered by their instability and indiscriminate targeting of other proteases (e.g., ADAM10) leading to high toxicity, thus paving the way for other ADAM17-based treatment modalities (e.g., antibody) to be explored in cancer models (Fridman et al, 2007; Richards et al, 2012; Rios-Doria et al, 2015; Ye et al, 2017).

The limited number of studies investigating ADAM17 in NSCLC has suggested that elevated ADAM17 expression correlates with numerous clinicopathological characteristics (e.g., tumor grade) and poor survival (Ni et al, 2013). Furthermore, in human NSCLC

cell lines, ADAM17 can modulate EGFR signaling either indirectly via Notch1 shedding and activation leading to increased EGFR expression, or directly via shedding of EGFR family ligands, the latter of which may contribute to radiotherapy resistance of NSCLC tumors (Zhou et al, 2006; Baumgart et al, 2010; Sharma et al, 2016). However, the role of ADAM17 in the predominant LAC subtype and its association with KRAS mutation status, as well as the involvement of other ADAM17 substrates in LAC, are unknown.

Here, we reveal that threonine phosphorylation (i.e., activation) of ADAM17 by p38 MAPK is a key feature of KRAS mutant LAC. In proof-of-concept preclinical studies involving genetically engineered and xenograft (human cell line and patient-derived) KRAS mutant LAC models, the genetic and therapeutic targeting of ADAM17, the latter with a new class of specific ADAM17 prodomain inhibitor (Wong et al, 2016), markedly suppressed tumor growth. Strikingly, the reliance by oncogenic KRAS for ADAM17 associated with the preferential processing of the ADAM17 substrate, IL-6R, to produce soluble (s) IL-6R that activated ERK1/2 MAPK via IL-6 trans-signaling (Garbers et al, 2018). Collectively, our discovery of a druggable ADAM17-sIL-6R axis in KRAS mutant LAC represents an attractive new strategy for the development of therapies for LAC and potentially other oncogenic KRAS-addicted cancers.

# Results

## Genetic reduction in ADAM17 suppresses the LAC phenotype of $Kras^{G12D}$ mice

To assess the role of ADAM17 in oncogenic Kras-driven LAC, we generated $Kras^{G12D}$ mice (expressing one mutant and one wild-type Kras allele) either heterozygous ($Kras^{G12D}$:$Adam17^{ex/+}$) or homozygous ($Kras^{G12D}$:$Adam17^{ex/ex}$) for the hypomorphic Adam17ex allele (Chalaris et al, 2010). Briefly, the engineered Adam17ex allele contains a new exon—starting with an in-frame translational stop codon and flanked by splice donor/acceptor sites between exons 11 and 12 of the endogenous gene encoding ADAM17—which in a homozygous state accounts for ~95% of the total ADAM17 mRNA pool, thus dramatically reducing wild-type ADAM17 protein levels while maintaining sufficient basal expression to maintain viability (Fig EV1A and B; Chalaris et al, 2010). At 6 weeks post-induction of the oncogenic $Kras^{G12D}$ allele by Ad-Cre inhalation, $Kras^{G12D}$:$Adam17^{ex/+}$ and $Kras^{G12D}$:$Adam17^{ex/ex}$ mice displayed a significant 30 and 68% reduction, respectively, in the area of lung parenchyma affected by diffuse atypical adenomatous hyperplasia (AAH) and sporadic adenocarcinoma in situ (AIS) lesions compared to $Kras^{G12D}$ littermates (Fig 1A and B). Similarly, the number of discreet lesions in $Kras^{G12D}$:$Adam17^{ex/+}$ and $Kras^{G12D}$:$Adam17^{ex/ex}$ lungs was markedly lower by 24 and 67%, respectively, compared to $Kras^{G12D}$ lungs (Fig 1C). In addition, numbers of epithelial alveolar type II (ATII) cells positive for the LAC marker thyroid transcription factor-1 (TTF-1) were significantly reduced throughout the whole lung, as well as in lesion areas, of $Kras^{G12D}$:$Adam17^{ex/+}$ and $Kras^{G12D}$:$Adam17^{ex/ex}$ mice compared to $Kras^{G12D}$ mice (Figs 1D and E, and EV1C). The suppressed LAC phenotype of $Kras^{G12D}$:$Adam17^{ex/ex}$ mice was also

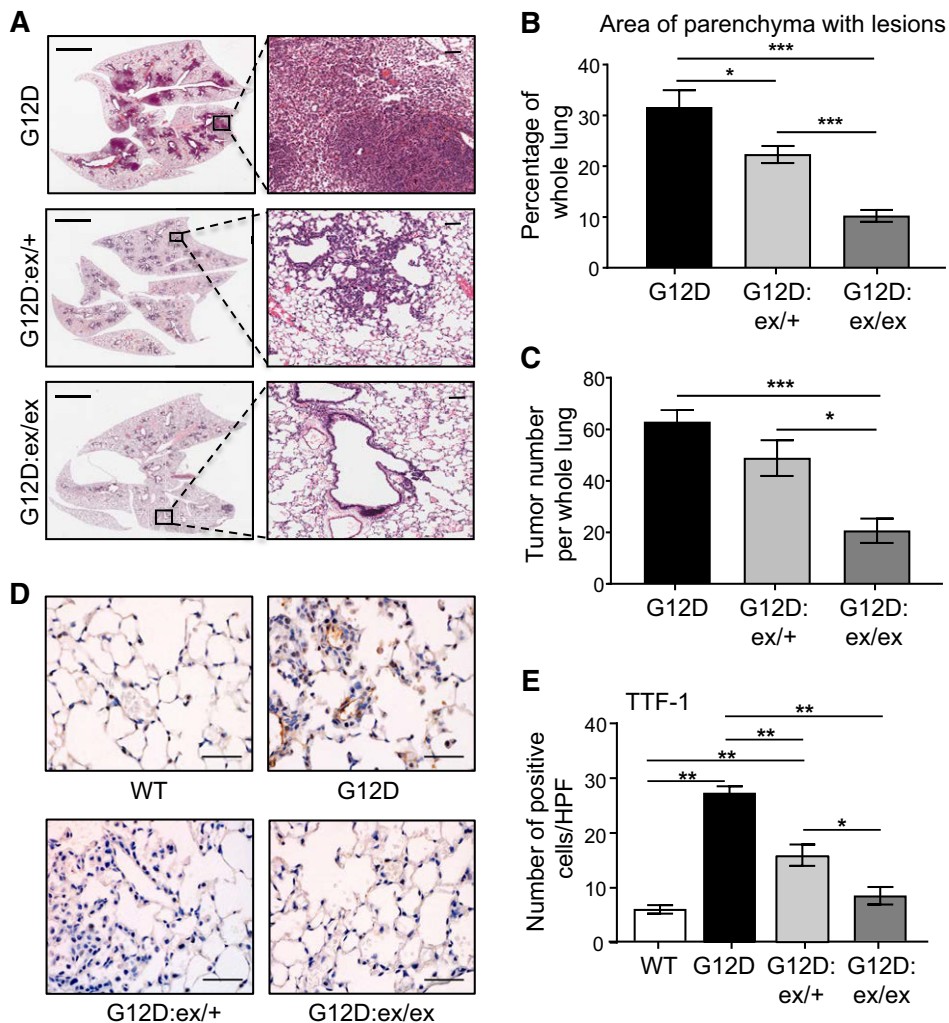

**Figure 1. ADAM17 deficiency abrogates oncogenic Kras-induced LAC.**

A    Representative low (left) and high (right) power images of H&E-stained lung sections from $Kras^{G12D}$, $Kras^{G12D}$:$Adam17^{ex/+}$, and $Kras^{G12D}$:$Adam17^{ex/ex}$ mice at 6 weeks post-Ad-Cre inhalation. Scale bars, 3 mm (left) and 300 μm (right).

B, C    Quantification of lung parenchyma area occupied by tumor lesions (B), and tumor incidence (C), per whole mouse lung in the indicated genotypes ($n = 6$ per genotype). *$P < 0.05$, ***$P < 0.001$, Student's $t$-test, mean ± SEM.

D    Representative images of TTF-1-stained lung sections from the indicated genotypes. Scale bar, 100 μm.

E    Quantification of TTF-1-positive cells/high-power field (HPF) in lungs of the indicated genotypes ($n = 6$ per genotype). *$P < 0.05$, **$P < 0.01$, Student's $t$-test, mean ± SEM.

Data information: Exact $P$ values are specified in Appendix Table S4.

observed at 12 weeks following oncogenic *Kras* activation, as evidenced by fewer numbers of more advanced AAH and AIS lesions, along with TTF-1-positive cells, compared to age-matched $Kras^{G12D}$ mice (Appendix Fig S1A–D). These findings suggest that ADAM17 plays a critical pro-tumorigenic role in $Kras^{G12D}$-induced LAC.

**ADAM17 promotes tumor cell proliferation and inflammation during $Kras^{G12D}$-induced LAC**

The suppressed LAC phenotype in $Kras^{G12D}$:$Adam17^{ex/ex}$ mice was associated with a significant reduction in the proliferative index of tumor-bearing $Kras^{G12D}$:$Adam17^{ex/ex}$ lungs, as measured by cellular reactivity to proliferating cell nuclear antigen (PCNA), compared to $Kras^{G12D}$ lungs (Figs 2A and B, and EV1D, Appendix Fig S1E and F). Furthermore, the reduced cellular proliferation in $Kras^{G12D}$:$Adam17^{ex/ex}$ lung lesions was accompanied by significantly lower expression levels of several cell cycle regulatory genes that are upregulated and implicated in human LAC (Fig 2C–E), among which *Myc* has been shown to cooperate with the oncogenic $Kras^{G12D}$ allele to promote the proliferation of early-stage lung tumor cells (Soucek *et al*, 2008; Chen *et al*, 2012; Xu *et al*, 2015; Gao & Wang, 2018).

In addition to its potent mitogenic signaling capacity within the lung epithelium, mutant Kras also promotes an inflamed tumor microenvironment associated with the production of various

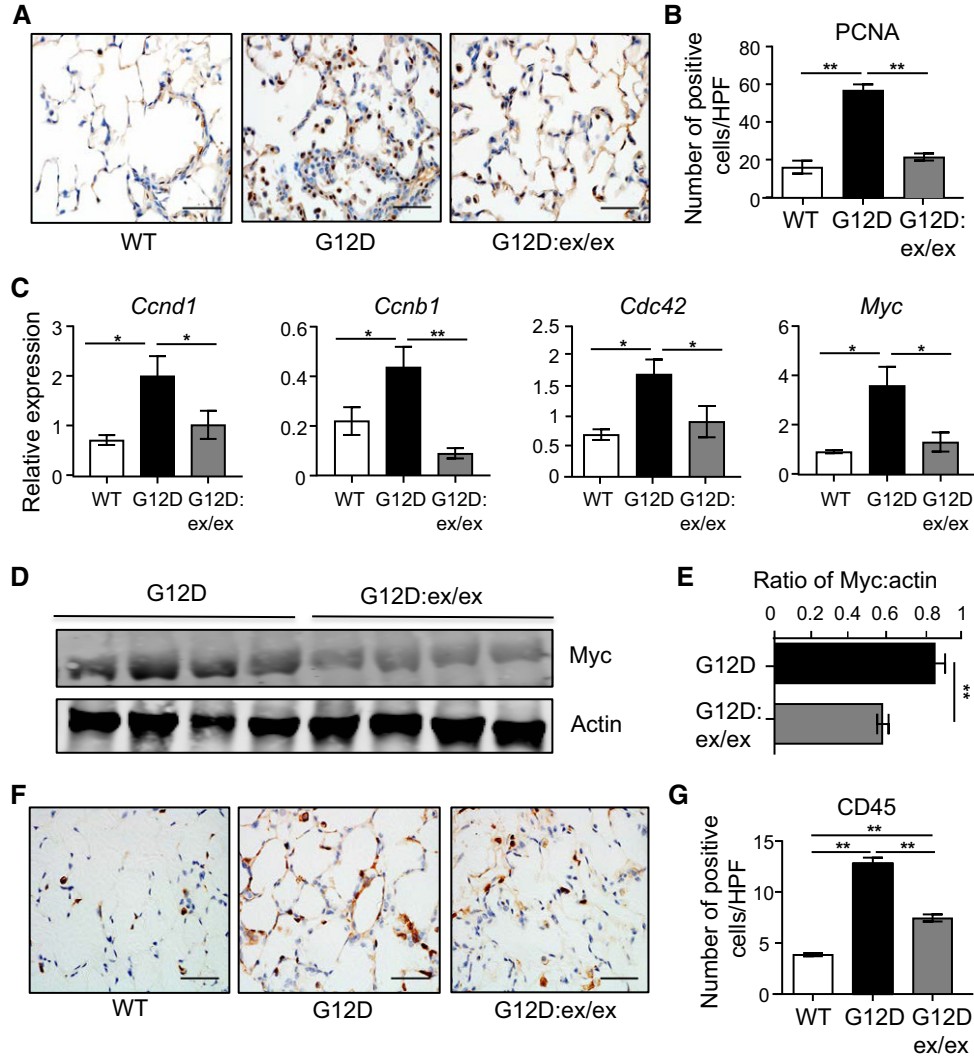

**Figure 2.  Reduced tumor cell proliferation and inflammation in *Kras*[G12D]:*Adam17*[ex/ex] mouse lungs.**

A    Representative images of PCNA-stained lung sections from *Kras*[WT], *Kras*[G12D], and *Kras*[G12D]:*Adam17*[ex/ex] mice at 6 weeks post-Ad-Cre (for *Kras*[G12D]) or vehicle (for *Kras*[WT]) inhalation. Scale bar, 100 μm.

B    Quantification of PCNA-positive cells/high-power field (HPF) in the indicated mouse lungs (n = 6 per genotype). **P < 0.01, Student's t-test, mean ± SEM.

C    qPCR expression analyses of cell cycle regulation genes (normalized against *18SrRNA*) in lungs from the indicated mice (n = 6 per genotype). *P < 0.05, Student's t-test, mean ± SEM.

D    Representative immunoblots of *Kras*[G12D] and *Kras*[G12D]:*Adam17*[ex/ex] lung lysates with the indicated antibodies. Each lane is an individual mouse sample.

E    Semi-quantitative densitometry of Myc protein levels (relative to actin) in lung lysates from (D). n = 6 per genotype. **P < 0.01, Student's t-test, mean ± SEM.

F, G    Representative images (F) and quantification of CD45-positive cells/HPF (G) in CD45-stained lung sections from the indicated genotypes at 6 weeks post-inhalations, mean ± SEM. In (F), scale bar, 100 μm. In (G), n = 6 per genotype. **P < 0.01, Student's t-test.

Data information: Exact P values are specified in Appendix Table S4.
Source data are available online for this figure.

cytokines (Ji *et al*, 2006; Sutherland *et al*, 2014). In this respect, immunohistochemical staining indicated that the increased numbers of CD45-positive total immune/inflammatory cells present in *Kras*[G12D] whole lung and lesions were significantly reduced compared to their *Kras*[G12D]:*Adam17*[ex/ex] counterparts (Figs 2F and G, and EV1E). Additional immunohistochemical analysis of immune cell subsets demonstrated that reduced numbers of total immune cells in *Kras*[G12D]:*Adam17*[ex/ex] lungs corresponded to F4/80-positive

macrophages, CD3-positive T cells, B220-positive B cells, and Ly6G-positive neutrophils (Fig EV2A–H). Furthermore, suppressed inflammation in *Kras*[G12D]:*Adam17*[ex/ex] lungs coincided with lower expression of numerous inflammatory mediators (Fig EV2I).

By contrast, the extent of apoptosis, as determined by active Caspase-3 immunostaining, in the lungs of *Kras*[G12D] and *Kras*[G12D]:*Adam17*[ex/ex] mice, was comparable (Fig EV2J and K). Similarly, immunostaining with the endothelial marker CD31, along with gene

expression profiling of an angiogenic gene signature (Brooks *et al*, 2016), indicated that the suppressed lung tumorigenesis in *Kras*[G12D]: *Adam17*[ex/ex] mice was independent of changes in angiogenesis (Fig EV2L–N). Collectively, these data strongly support the notion that ADAM17 promotes *Kras*[G12D]-dependent LAC by supporting tumor cell proliferation and inflammation.

### ADAM17 is expressed in tumor-initiating epithelial cell types in the lungs of *Kras*[G12D] mice, with no tumor-promoting role for the ADAM17-expressing myeloid lineage

The requirement of ADAM17 for the hyper-proliferative and inflammatory phenotype of mutant *Kras*-driven LAC raises the question as to whether ADAM17 expression in immune cells, as well as in the lung epithelium from which oncogenic Kras[G12D] signaling emanates, contributes to LAC. Therefore, we investigated the pulmonary cellular compartments of *Kras*[G12D] mice that express ADAM17 by performing immunofluorescence staining with an antibody recognizing both pro- and mature forms of ADAM17. In tumor-bearing regions of *Kras*[G12D] mouse lungs, ADAM17 staining was mainly detected among different epithelial cell types—club (previously known as Clara; CC10-positive), ATII (SPC-positive), and ATI (podoplanin-positive) cells—CD45-positive immune cells and, to a lesser extent, CD31-positive endothelial cells (Fig 3A–F). Of note, ATII and club cells have been identified as the cells of origin of mutant Kras-driven LAC (Sutherland *et al*, 2014), thus suggesting that ADAM17 may contribute to tumor initiation in the lung.

The expression of ADAM17 in immune cells, along with the capacity of *Kras*[G12D]-expressing myeloid cells to promote lung tumorigenesis (Kamata *et al*, 2017), led us to investigate whether ADAM17 expression in myeloid cells also contributed to Kras-driven LAC. However, reconstitution of the hematopoietic compartment of irradiated *Kras*[G12D] recipient mice with donor bone marrow from *Kras*[G12D]:*Adam17*[ex/ex] mice did not alleviate the LAC phenotype of the recipient mice (Fig 3G and H). Taken together, these observations support the notion that ADAM17 expression in the pulmonary epithelium, but not the hematopoietic-derived myeloid lineage, plays a major role in promoting mutant Kras-induced LAC.

### Augmented p38 MAPK-induced threonine phosphorylation of ADAM17 in *Kras*[G12D]-induced LAC

We next explored whether the requirement for ADAM17 in oncogenic Kras-induced LAC was associated with upregulated expression of mature ADAM17 protein. However, expression levels of ADAM17 mRNA and pro (120 kDa)- and mature (90 kDa) protein forms were comparable in the lungs of *Kras*[G12D] mice and tumor-free *Kras*[WT] controls (Appendix Fig S2A–C). In addition, the gene expression of numerous upstream regulators of ADAM17 maturation and/or activation was unchanged in *Kras*[G12D] mouse lungs (Appendix Fig S2D). By contrast, immunohistochemistry and immunoblot analyses indicated that levels of threonine (Thr[735]) phosphorylated (p) ADAM17, which is a hallmark of its activation (Zunke & Rose-John, 2017), were significantly upregulated in *Kras*[G12D] mouse lungs (Fig 4A–D). Since threonine phosphorylation of the ADAM17 cytoplasmic domain by p38 and ERK1/2 MAPKs activates ADAM17-mediated shedding (Soond *et al*, 2005; Killock

& Ivetić, 2010), we investigated the phosphorylation (i.e., activation) status of these MAPKs. Consistent with previous observations for low and/or unaltered pERK1/2 MAPK levels downstream of oncogenic Kras, the mechanistic basis for which remains unresolved (Tuveson *et al*, 2004; Cicchini *et al*, 2017), pERK1/2 levels were unchanged in *Kras*[G12D] mouse lungs (Fig 4E and F). By contrast, we observed a significant increase in pp38 MAPK levels in the lungs of *Kras*[G12D] mice compared to *Kras*[WT] controls (Fig 4E and F).

To verify the selective requirement of p38 MAPK for ADAM17 Thr[735] phosphorylation, we assessed pADAM17 levels in the lungs of *Kras*[G12D] mice administered with pharmacological inhibitors against p38 (SB203580) and ERK1/2 (U0126) MAPKs. While SB203580 significantly suppressed pADAM17-positive cell numbers in lesion areas of *Kras*[G12D] lungs, U0126 had no effect (Fig 4G and H). Importantly, these observations were supported in *KRAS* mutant human LAC cells A549 and NCI-H23, whereby treatment with SB203580 to suppress p38 MAPK activity (verified by lower phosphorylated levels of HSP27, a p38 target) reduced pADAM17 protein levels (Fig 4I), along with cell growth (Fig 4J and K). Taken together, these data support the notion that ADAM17 threonine phosphorylation, via p38 MAPK, augments *Kras*[G12D]-dependent LAC.

### ADAM17 preferentially processes the soluble IL-6R, which potentiates IL-6 trans-signaling via the ERK1/2 MAPK pathway, in *Kras*[G12D]-induced LAC

Since ADAM17 sheds a large range of bioactive substrates that activate numerous intracellular signaling cascades associated with oncogenic Kras (e.g., STAT3, AKT, and MAPK), we investigated the pathways that were specifically influenced by ADAM17 in *Kras*[G12D]-induced LAC. Immunoblot analysis revealed a striking reduction in ERK1/2 MAPK phosphorylation in lungs of *Kras*[G12D]:*Adam17*[ex/ex] versus parental *Kras*[G12D] mice, whereas the phosphorylation status of other intracellular signaling mediators (STAT3, AKT) remained unchanged upon modulating either Kras activation or ADAM17 expression levels in the lung (Fig 4L). In support of this finding, immunohistochemistry indicated a significant reduction in pERK1/2 MAPK-positive cell numbers in *Kras*[G12D]:*Adam17*[ex/ex] mouse whole lung and lung lesions (Figs 4M and N, and EV1F, Appendix Fig S1G and H). The reduction in pERK1/2 levels was limited to the lungs, since pERK1/2 levels were unchanged in other tissues of *Kras*[G12D]: *Adam17*[ex/ex] mice (Fig EV3A).

We next performed ELISA on serum or immunoblotting on lung tissue lysates from *Kras*[G12D] versus *Kras*[G12D]:*Adam17*[ex/ex] mice, along with control *Kras*[WT] mice, to investigate whether processed downstream substrates of ADAM17 known to signal via ERK1/2 MAPK, namely sIL-6R, cleaved Notch1, and EGFR family ligands (e.g., TGFα and neuregulin-1) (Zhou *et al*, 2006; Baumgart *et al*, 2010; Brooks *et al*, 2016; Zunke & Rose-John, 2017), were associated with ADAM17-mediated oncogenic Kras-induced LAC. Somewhat surprisingly, among these shed ADAM17 targets, only the sIL-6R was significantly downregulated in serum from *Kras*[G12D]: *Adam17*[ex/ex] compared to *Kras*[G12D] mice (Fig 5A and B). We verified that reduced sIL-6R levels were not due to lower mRNA and protein expression of membrane-bound full-length IL-6R in *Kras*[G12D]:*Adam17*[ex/ex] mouse lungs (Fig EV3B and C). This

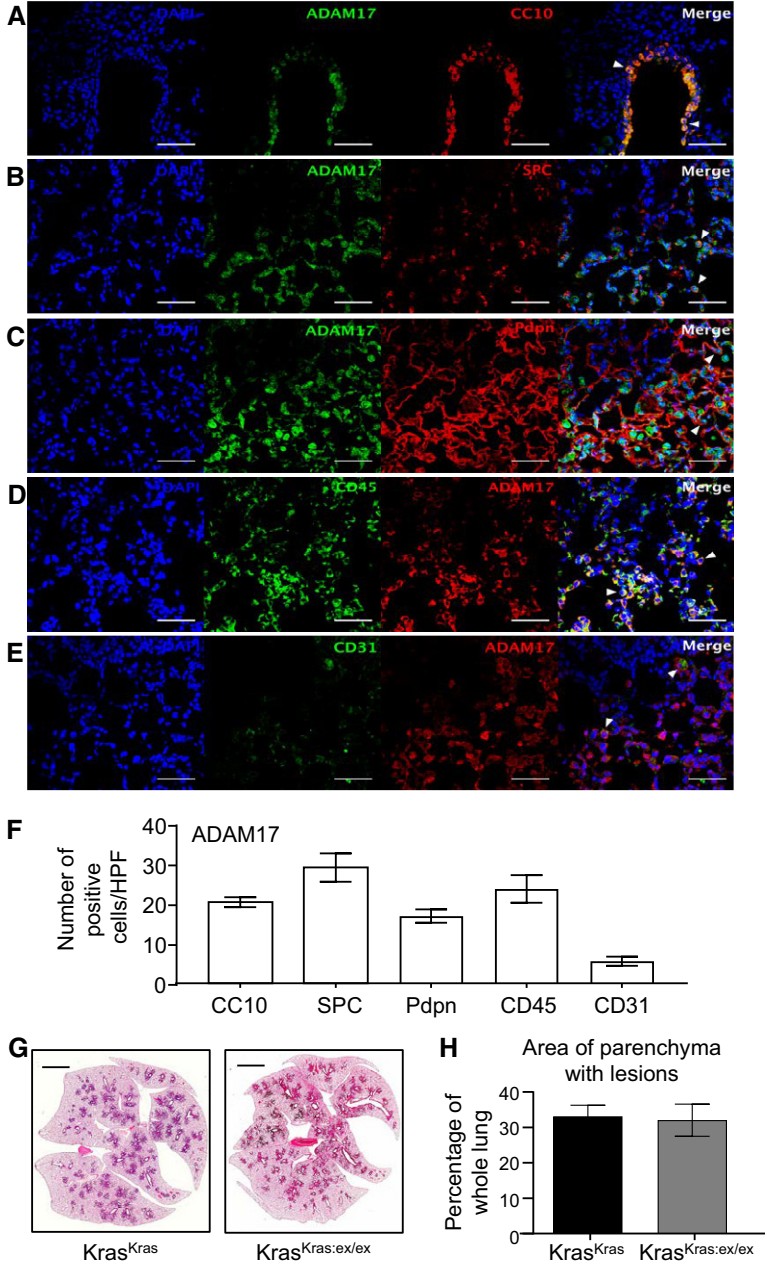

**Figure 3. Cellular expression pattern of ADAM17 in lungs of tumor-bearing mice.**

A–E   Representative immunofluorescence images of lung sections of *Kras*[G12D] mice (6 weeks post-Ad-Cre inhalation) co-stained for ADAM17 and markers for club cells (CC10) (A), alveolar type II (surfactant protein-C, SPC) (B), and type I (podoplanin, Pdpn) (C) cells, total immune cells (CD45) (D), and endothelial cells (CD31) (E). DAPI nuclear staining is blue. Scale bars, 100 μm. Arrowheads in merged images indicate dual-positive ADAM17-expressing cells.

F   Quantification of immunofluorescence staining from (A-E) (*n* = 5 mice per stain), mean ± SEM.

G   Representative images of H&E-stained lung sections from *Kras*[G12D] recipient mice reconstituted with *Kras*[G12D] (Kras[Kras]) or *Kras*[G12D]:*Adam17*[ex/ex] bone marrow (Kras[Kras:ex/ex]). Scale bar, 3 mm.

H   Quantification of lung parenchyma area occupied by tumor lesions in the indicated bone marrow chimeras (*n* = 6 mice per group), mean ± SEM.

Data information: Exact *P* values are specified in Appendix Table S4.

reduction in sIL-6R levels was specific, since expression levels of other processed ADAM17 substrates, such as those for TNFα and TGFα, along with Notch1-regulated genes, and phosphorylated ErbB3 and EGFR receptors (downstream read-out for EGFR family ligand production), were comparable in serum and/or lung tissues of *Kras*[G12D] and *Kras*[G12D]:*Adam17*[ex/ex] mice (Figs 5A and B, and EV3D). Therefore, these data support the biased shedding of sIL-6R by ADAM17 in *Kras*[G12D] mice.

We further investigated the specific requirement for the IL-6R substrate by ADAM17 in mutant *Kras*-induced LAC by performing

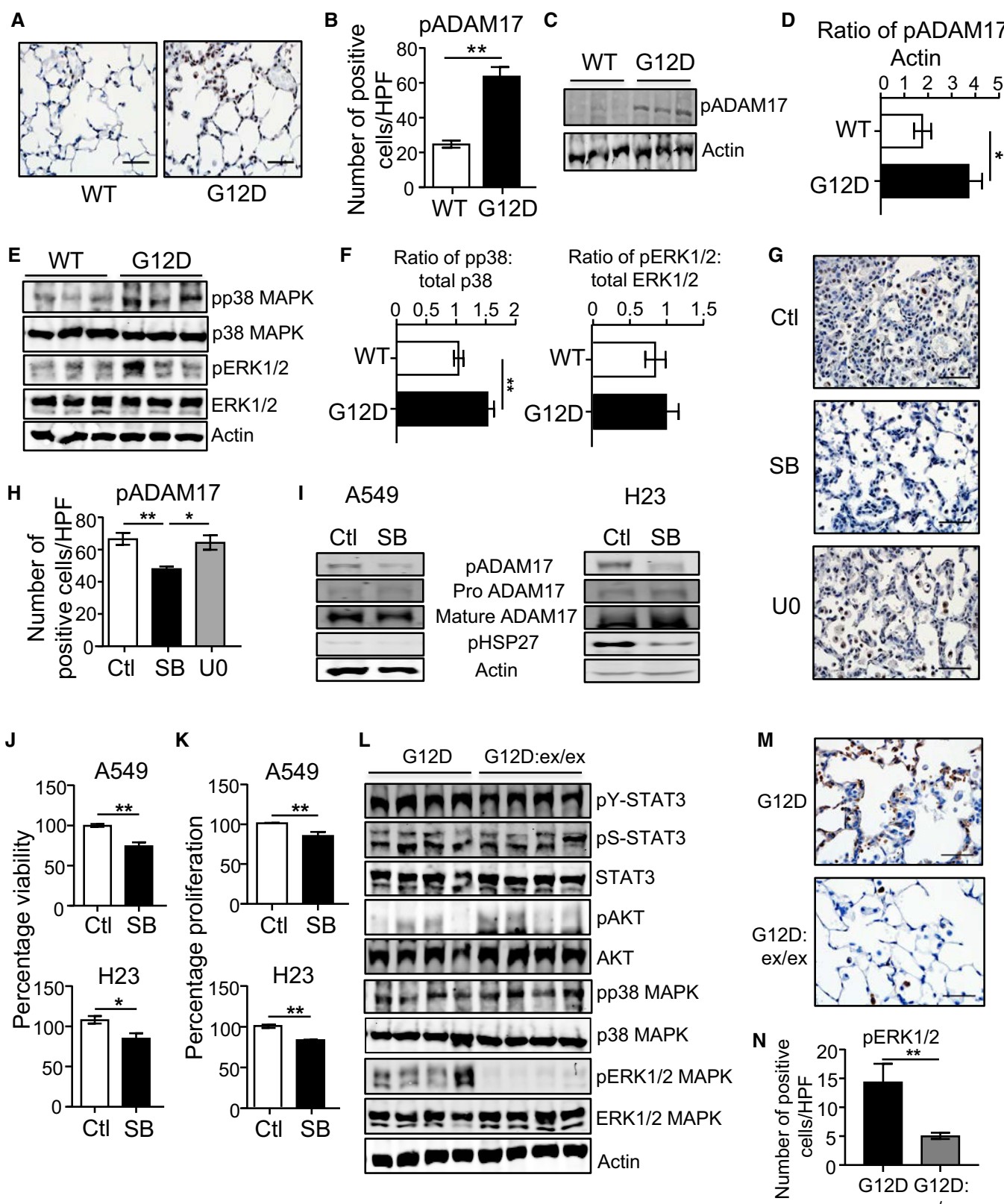

**Figure 4.**

**Figure 4.  Lung tumorigenesis in *Kras*$^{G12D}$ mice is associated with upstream modulation of ADAM17 Thr$^{735}$ phosphorylation by p38 MAPK and downstream ADAM17-mediated ERK1/2 MAPK signaling.**

A   Representative images of pADAM17-stained lung sections from *Kras*$^{WT}$ and *Kras*$^{G12D}$ (6 weeks post-inhalation) mice. Scale bar, 100 μm.

B   Quantification of pADAM17-positive cells/high-power field (HPF) in lungs of *Kras*$^{WT}$ and *Kras*$^{G12D}$ mice ($n$ = 6 per genotype). **$P$ < 0.01, Student's $t$-test, mean ± SEM.

C   Representative immunoblots of individual *Kras*$^{WT}$ and *Kras*$^{G12D}$ mouse lung lysates with the indicated antibodies.

D   Semi-quantitative densitometry of pADAM17 in lung lysates ($n$ = 6 per genotype). *$P$ < 0.05, Student's $t$-test, mean ± SEM.

E   Representative immunoblots of individual *Kras*$^{WT}$ and *Kras*$^{G12D}$ lung lysates with the indicated antibodies.

F   Semi-quantitative densitometry of pp38 and pERK1/2 MAPK protein levels in *Kras*$^{WT}$ and *Kras*$^{G12D}$ lung lysates ($n$ = 6 per genotype). **$P$ < 0.01, Student's $t$-test, mean ± SEM.

G   Representative images of pADAM17-stained lung sections from *Kras*$^{G12D}$ mice injected with a single dose of vehicle (control), SB203580 p38 MAPK or U0126 ERK1/2 inhibitors. Scale bar, 100 μm.

H   Quantification of pADAM17-positive cells/HPF in lungs of the indicated groups ($n$ = 5 per group). *$P$ < 0.05, **$P$ < 0.01, Student's $t$-test, mean ± SEM.

I   Immunoblots with the indicated antibodies on lysates from A549 and NIH-H23 LAC cell lines treated with SB203580 for 24 h.

J, K   MTT viability (J) and ATP proliferation (K) assays of A549 and NCI-H23 LAC cells treated with SB203580. Data are from three independent experiments performed in triplicate. *$P$ < 0.05, **$P$ < 0.01, Student's $t$-test, mean ± SEM.

L   Representative immunoblots of individual *Kras*$^{G12D}$ and *Kras*$^{G12D}$:*Adam17*$^{ex/ex}$ lung lysates with the indicated antibodies. For the actin blot, lanes 1–3 are reproduced in lanes 4–6 of the actin blot in Appendix Fig S2B. For the pp38/p38 MAPK blots, lanes 1–3 are reproduced in lanes 4–6 of the corresponding blots in panel (E).

M   Representative images of pERK1/2-stained lung sections from the indicated genotypes. Scale bars, 100 μm.

N   Quantification of pERK1/2-positive cells/high-power field (HPF) in lungs of the indicated genotypes ($n$ = 6 per genotype). **$P$ < 0.01, Student's $t$-test, mean ± SEM.

Data information: Exact $P$ values are specified in Appendix Table S4.
Source data are available online for this figure.

immunofluorescence of lung tissue from *Kras*$^{G12D}$ mice. IL-6R presented a similar and wide-spread co-localization pattern of staining among ATII and club epithelial cells, immune cells, and endothelial cells to that of ADAM17 (Fig EV3E–H; see also Fig 3A–E). IL-6R also displayed pronounced co-localization with ADAM17 throughout *Kras*$^{G12D}$ mouse lungs, which contrasted the limited expression and co-localization of other ADAM17 substrates, namely TGFα, neuregulin-1 (Nrg1), EGF, and Notch1 (Fig 5C). Moreover, the targeted blockade of IL-6 trans-signaling with neutralizing anti-IL-6R antibodies 1F7 and 25F10, which suppress *Kras*$^{G12D}$-induced LAC (Brooks *et al*, 2016), also markedly reduced pERK1/2-positive cell numbers in *Kras*$^{G12D}$ lungs (Fig 5D and E). In addition, consistent with a role for oncogenic Kras-mediated signaling events emanating from the lung epithelium, stimulation of IL-6 trans-signaling with the potent agonist Hyper-IL-6 specifically upregulated (by 65%) ERK1/2 MAPK signaling in epithelial cells, but not immune cells, isolated from mouse lungs harboring the activated *Kras*$^{G12D}$ allele (Fig 5F). Collectively, these observations invoke the existence of a novel ADAM17-sIL-6R-ERK MAPK signaling axis that facilitates oncogenic Kras-induced LAC.

## Upregulated ADAM17 activation is a common feature of human *KRAS* mutant LAC and promotes tumor growth *in vivo*

To translate the clinical utility of our findings in *Kras*$^{G12D}$ mice, we initially investigated the activation status of ADAM17 in human LAC patients either wild-type or mutant for *KRAS*. In support of our *in vivo* data, significantly increased numbers (75%) of pADAM17-expressing cells were detected in lung tumor sections of human LAC patients compared to cancer-free controls (Fig 6A and B). Moreover, in tumor biopsies, pADAM17 staining was primarily observed in epithelial (SPC-positive) cells, rather than immune (CD45-positive) cells (Fig EV4A). Cell numbers positive for pADAM17 or pERK1/2 staining were also significantly elevated in LAC patients stratified for mutant *KRAS* compared to wild-type *KRAS* (Fig 6C). In addition, using serial lung sections, we observed a significant positive correlation in the increased numbers of positive cells for pADAM17 with

those for pERK1/2, as well as with those for pp38, in LAC patients (Fig 6D and E). The increased pADAM17 activity in LAC patients was not a consequence of elevated ADAM17 expression, since ADAM17 mRNA and protein (pro/mature) levels were unchanged in tumor versus non-tumor tissues in LAC patient cohorts (Fig EV4B and C). Similarly, the expression levels of various other ADAM17 processed substrates and pEGFR were not elevated in LAC versus cancer-free lung lysates (Fig EV4C). Consistent with these clinical data, cellular levels of pADAM17, pERK1/2, and pp38 MAPK, along with culture supernatant levels of sIL-6R, were augmented in *KRAS* mutant human LAC cell lines A549 and NIH-H23 compared to *KRAS* wild-type human bronchial epithelial cells (HBEC) and NIH-H2228 LAC cells (Figs 6F–H and EV4D).

To evaluate the biological significance of elevated ADAM17 activity in human *KRAS* mutant LAC cells, we employed CRISPR/Cas9-mediated gene editing to ablate ADAM17 expression in A549 and NIH-H23 cells (Fig EV4E and F). Notably, cell viability and proliferation were significantly impaired in both ADAM17-deficient cell lines compared to ADAM17-expressing non-targeting control cell lines and coincided with a marked reduction in sIL-6R production (Figs 6I–K and EV5A–C). Furthermore, the growth of ADAM17-deficient A549 tumor xenografts in NSG mice was significantly impaired compared to ADAM17-expressing control A549 xenografts, as demonstrated by ~4-fold and ~5-fold reductions in final (day 21) tumor volumes and weights, respectively (Fig 6L–N). In addition, compared to NSG mice harboring control A549 xenografts, serum levels of human (i.e., tumor-derived) sIL-6R, along with cellular staining for pADAM17 and pERK1/2 within tumors, were substantially reduced in mice with ADAM17-deficient A549 tumor xenografts (Fig 6O and P). The biased processing of IL-6R (to produce sIL-6R) by ADAM17 in *KRAS* mutant LAC was supported by the observation that processed (i.e., mature) forms of other ADAM17 substrates (e.g., neuregulin-1 and TGFα) were comparable between control A549 and ADAM17-deficient A549 xenografts (Fig 6Q). In addition, serum levels of other shed substrates (e.g., TNFα) were undetected in mice bearing control A549 xenografts (not shown). Collectively, the above

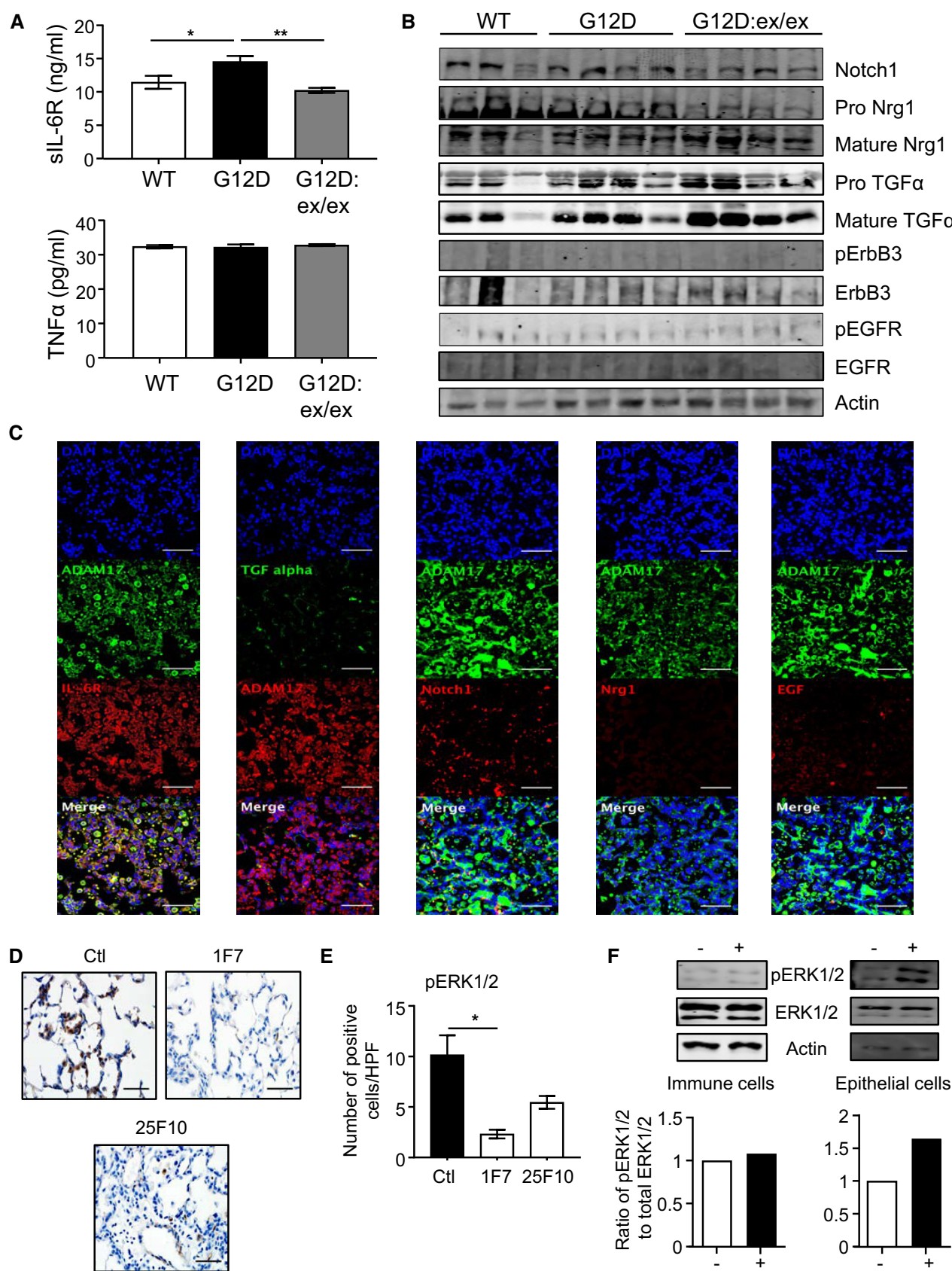

**Figure 5.**

◀

**Figure 5. In *Kras*[G12D]-induced LAC, ADAM17-mediated processing of sIL-6R is selectively impaired, and specific targeting of sIL-6R reduces ERK MAPK activation.**

A    ELISA of sIL-6R and TNFα protein levels in serum from *Kras*[G12D] and *Kras*[G12D]:*Adam17*[ex/ex] mice at 6 weeks post-Ad-Cre inhalation, along with control *Kras*[WT] mice at 6 weeks post-vehicle inhalation (n = 6 per genotype). *P < 0.05, **P < 0.01, Student's t-test, mean ± SEM.

B    Immunoblots of representative individual *Kras*[WT], *Kras*[G12D], and *Kras*[G12D]:*Adam17*[ex/ex] lung lysates with the indicated antibodies.

C    Representative immunofluorescence staining of *Kras*[G12D] mouse lung sections (n = 5 mice) for ADAM17 and either (panels from left to right) IL-6R, TGFα, Notch1, Nrg1, and EGF. Sections were counterstained with DAPI (blue). Scale bars, 100 μm.

D    Representative images of pERK1/2-stained lung sections from *Kras*[G12D] mice treated with either vehicle (control, Ctl) or anti-IL-6R antibodies 1F7 or 25F10 over 6 weeks post-Ad-Cre inhalation. Scale bar, 100 μm.

E    Quantification of pERK1/2-positive cells/high-power field (HPF) in lungs of the indicated groups (n = 5 per group). *P < 0.05, Student's t-test, mean ± SEM.

F    Representative immunoblots with the indicated antibodies on whole cell lysates from primary immune and epithelial cells isolated from the lungs of mice (n = 3) harboring the oncogenic *Kras*[G12D] allele that were stimulated with either PBS (−) or 100 ng/ml Hyper-IL-6 (+). Graphs depict semi-quantitative densitometry of pERK1/2 MAPK protein levels in stimulated immune and epithelial cells (n = 2 experiments).

Data information: Exact P values are specified in Appendix Table S4.
Source data are available online for this figure.

data strongly suggest that pADAM17, coupled with downstream activation of the sIL-6R-ERK1/2 MAPK axis, supports tumor growth in *KRAS* mutant human LAC.

## Targeted therapy with the specific ADAM17 prodomain inhibitor, A17pro, suppresses *KRAS* mutant tumor growth in *Kras*[G12D] and patient-derived xenograft LAC models

Although ADAM17 SMIs show anti-cancer efficacy in numerous preclinical models, these inhibitors are highly toxic due to the non-specific targeting of other metalloproteinases (e.g., ADAM10), thus highlighting the need for highly selective ADAM17 inhibitors with an alternate mode of action (Moss & Minond, 2017). Recently, a recombinant protein of the auto-inhibitory prodomain of ADAM17, A17pro, has been developed which specifically abrogates the sheddase activity of ADAM17, but not ADAM10, and shows marked therapeutic efficacy without toxicity in numerous *in vivo* inflammatory disease models (Kefaloyianni *et al*, 2016; Wong *et al*, 2016). Notably, A17pro treatment of *KRAS* mutant human A549 and NIH-H23 LAC cells significantly reduced cell viability and proliferation, along with sIL-6R levels in culture supernatants, compared to control vehicle-treated cells (Fig EV5D–F), thus mimicking ADAM17-deficient cells (Fig 6I–K). Similarly, the formation of 3-dimensional spheroid aggregates was dramatically diminished in A549 cells either deficient in ADAM17 or treated with A17pro, suggesting that ADAM17 can also impact on the proliferative capacity of cancer stem (i.e., initiating) cells in LAC (Figs EV5G and 5H). Furthermore, these *in vitro* activities of A17pro were comparable to those of the dual ADAM10/17 inhibitor, GW280264X (Fig EV5I–K).

We next assessed the *in vivo* activity of A17pro in the *Kras*[G12D] LAC model. Compared to vehicle treatment, A17pro significantly suppressed the LAC phenotype of *Kras*[G12D] mice (Fig 7A and B), which was associated with reduced numbers of TTF-1-, PCNA-, and pERK1/2-positive cells in the lung (Fig 7C–H), as well as lower serum levels of sIL-6R (Fig 7I). Importantly, the anti-cancer activity of A17pro was verified in a human *KRAS* mutant LAC patient-derived xenograft expressing high levels of pADAM17. Here, A17pro also significantly suppressed tumor growth, which was again associated with lower numbers of PCNA- and pERK1/2-positive cells in the treated tumor xenografts, as well as reduced levels of serum human sIL-6R (Fig 7J–O, Appendix Fig S3). Collectively, these

observations demonstrate the potent inhibitory effect of A17pro on *KRAS* mutant LAC, comparable to that observed in the *Kras*[G12D] LAC model.

# Discussion

Activating point mutations in *KRAS* are among the most prevalent genetic alterations in cancer, with highest incidences specific for certain epithelial (e.g., lung, pancreatic, and colorectal) and hematological (e.g., acute myeloid leukemia and multiple myeloma) malignancies (Prior *et al*, 2012). In addition to being a potent driver event in numerous cancers, the clinical utility of oncogenic mutation of *KRAS* has been reported in the context of prognosis of poor survival outcomes, diagnosis of malignancy, and prediction of response and/ or acquired resistance to both chemotherapy and targeted therapies against EGFR (i.e., TKIs) and, very recently, PD-L1-based checkpoint inhibition (Hames *et al*, 2016; Dong *et al*, 2017; Hirsch *et al*, 2017; Zhuang *et al*, 2017; Román *et al*, 2018). However, the clinical significance of *KRAS* as a potential prognostic/predictive biomarker is tempered by contrasting findings, for instance in NSCLC, which most likely reflects the high degree of genetic and molecular heterogeneity among patient tumors (Shepherd *et al*, 2013). Despite the importance of mutant *KRAS* in oncology, the development and clinical implementation of specific KRAS inhibitors for targeted therapy have been unsuccessful to date. In this respect, a spate of recent high profile reports has described a new generation of SMIs that trap mutant KRAS (G12C) in a GDP-bound inactive state, with one compound in particular (ARS-1620) displaying robust anti-tumor activity in numerous NSCLC xenograft models harboring the *KRAS*[G12C] allele (Patricelli *et al*, 2016; Janes *et al*, 2018). While the broader clinical activity and safety of such compounds remain to be evaluated, they nonetheless challenge the long-held dogma that KRAS is "undruggable" (Cox *et al*, 2014).

The absence of effective targeted therapies for LAC (and other cancers) driven by KRAS-activating mutations places a great emphasis on identifying cooperating factors and/or downstream effectors which mediate the oncogenic activity of mutant *KRAS* and with the potential to guide the development of indirect strategies to target oncogenic KRAS. This need is further emphasized by the observation that in *KRAS* mutant NSCLC cell lines, RNAi-mediated knockdown of KRAS only partially inhibits *in vivo* tumor growth,

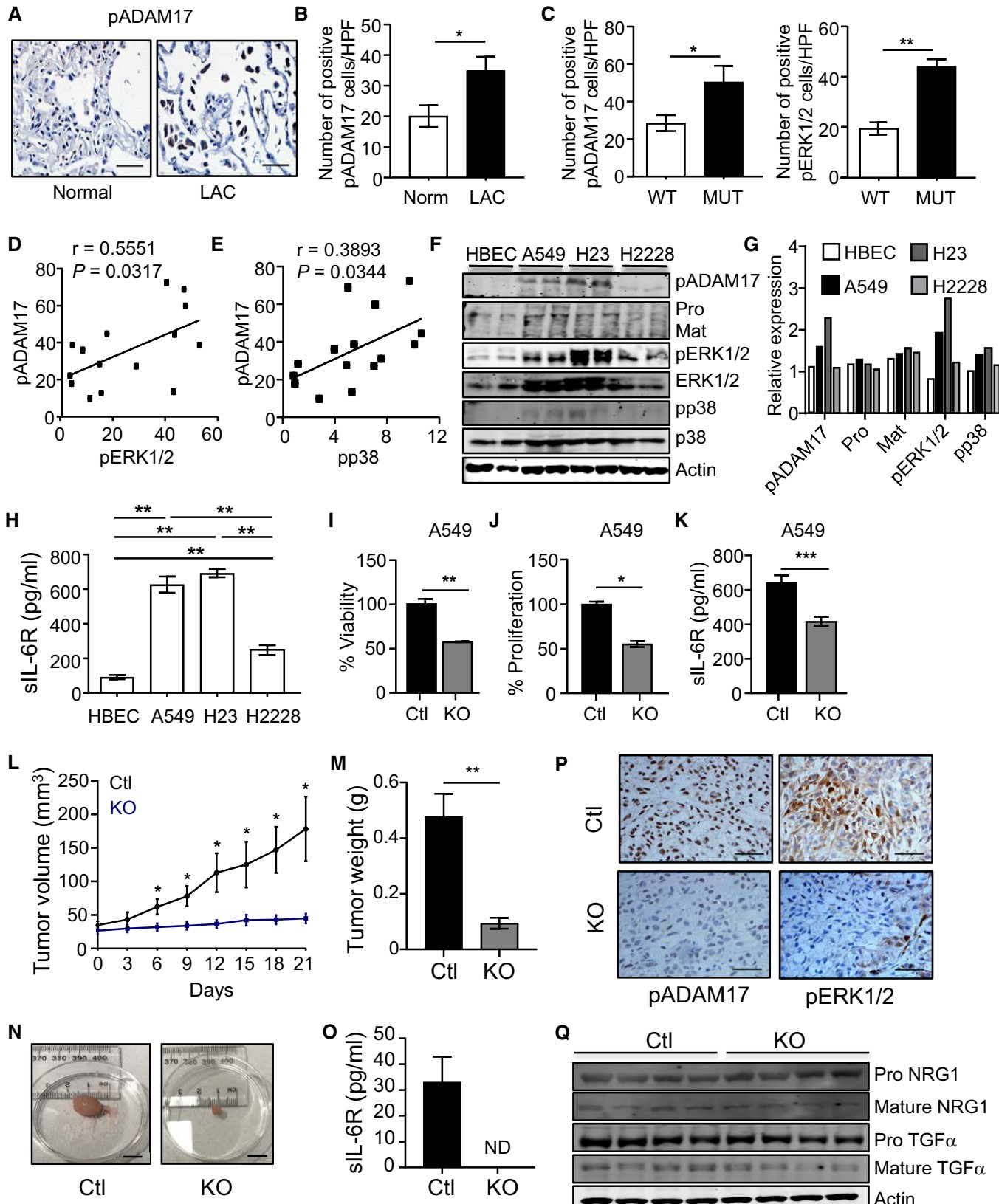

**Figure 6.**

◄

**Figure 6.  Enhanced activation of the ADAM17-sIL-6R-ERK1/2 MAPK axis promotes tumor growth in human *KRAS* mutant LAC.**

A       Representative images of pADAM17-stained lung sections from normal (cancer-free) control and LAC patients. Scale bar, 100 μm.

B, C    Quantification of pADAM17-positive cells (B and C) and pERK1/2 MAPK-positive cells (C) in (B) normal (cancer-free) control individuals ($n = 8$) and LAC patients ($n = 18$), and (C) LAC patients either *KRAS* wild-type (WT, $n = 13$) or mutant (MUT, $n = 5$). *$P < 0.05$, **$P < 0.01$, Student's $t$-test, mean ± SEM.

D, E    Linear regression analyses of pADAM17 with pERK1/2 (D) and pp38 (E) MAPK staining in serial lung sections from LAC *KRAS* wild-type and mutant patients. r, Pearson correlation coefficient.

F       Immunoblots of lysates from normal human bronchial epithelial cells (HBEC) and LAC cell lines A549, NIH-H23, and NIH-H2228 with the indicated antibodies. Shown are lysates from 2 separate passages per cell line. Pro, proADAM17; Mat, mature ADAM17.

G       Densitometric semi-quantification of immunoblots from (F) showing the relative expression levels of pADAM17 (relative to actin), pro- and mature forms of ADAM17 (relative to actin), and pERK1/2 and pp38 MAPK (relative to their total counterparts) protein levels.

H       ELISA of sIL-6R protein levels in culture supernatants from matching cells in (F) cultured over 24 h. Data are from 4 independent experiments. **$P < 0.01$, Student's $t$-test, mean ± SEM.

I, J    MTT viability (I) and ATP proliferation (J) assays of A549 cells transduced with non-targeted control sgRNA (Ctl) and ADAM17 sgRNA (knockout, KO). Data are from three independent experiments performed in triplicate. *$P < 0.05$, **$P < 0.01$, Student's $t$-test, mean ± SEM.

K       ELISA of sIL-6R protein levels in culture supernatants from matching A549 cells (I and J) cultured over 24 h. Data are from three independent experiments. ***$P < 0.001$, Student's $t$-test, mean ± SEM.

L       Tumor volume (mm³) of KO and Ctl A549 xenografts assessed every 3 days starting at 1 week post-injection of $10^6$ cells into NSG mice ($n = 5$ per group). *$P < 0.05$, Student's $t$-test, mean ± SEM.

M, N    Weight (grams) (M) and representative images (N) of tumor xenografts from (L). Scale bar, 1 cm. In (M), $n = 5$ per group. **$P < 0.01$, Student's $t$-test, mean ± SEM.

O       ELISA of human sIL-6R protein levels in serum of the indicated mice ($n = 5$ per group). Levels of sIL-6R are below detection in serum from KO xenograft mice. ND, not detected, mean ± SEM.

P       Representative images of pADAM17 and pERK1/2 MAPK immunohistochemical staining on KO and Ctl A549 xenograft tumors. Scale bars, 100 μm.

Q       Immunoblots of lysates from individual A549 KO and Ctl xenograft tumors with the indicated antibodies.

Data information: Exact $P$ values are specified in Appendix Table S4.

Source data are available online for this figure.

suggesting that targeting oncogenic KRAS in combination with other effectors will most likely provide enhanced anti-cancer efficacy compared to targeting KRAS alone (Sunaga *et al*, 2011). Here, we provide evidence that ADAM17 serves as a *bona fide* therapeutic target in mutant *KRAS*-driven LAC. Specifically, the highly selective blockade of ADAM17 using either genetic or inhibitor-based approaches in preclinical genetically engineered and xenograft model systems suppresses the oncogenic actions of mutant *KRAS* in LAC. In this regard, a key finding of our study was the robust anti-cancer activity of a novel class of highly selective ADAM17 inhibitor, A17pro, modeled on its auto-inhibitory prodomain (Wong *et al*, 2016). Importantly, A17pro is cross-species reactive (i.e., mouse and human) and thus provides an advantage over human-specific ADAM17 antibodies which cannot be evaluated for efficacy in genetically engineered mouse models of cancer (Richards *et al*, 2012).

In light of the promising preclinical activity of A17pro in the oncogenic setting of *KRAS* mutant LAC, we note that previous studies based on the use of non-specific SMIs (e.g., hydroxamates) or RNA interference-based approaches (siRNA, shRNA) in NSCLC cell lines have suggested that ADAM17 may have anti-cancer activity in lung cancer (Zhou *et al*, 2006; Baumgart *et al*, 2010; Lv *et al*, 2014; Sharma *et al*, 2016). However, in contrast to our current study, the clinical utility of ADAM17 specifically in mutant *KRAS* LAC (which accounts for ~15% of all lung cancers) was not investigated. Another limitation of these cell line-based studies, from a therapeutic viewpoint, was the use of targeting approaches whose translation to the clinic in oncology has not progressed due to a combination of challenges involving safety, efficacy, target selection, and delivery platforms (Pecot *et al*, 2011; Kamata *et al*, 2017).

Considering the high degree of molecular diversity observed within *KRAS* mutant tumors (including LAC), our current study addresses the unmet clinical need to identify novel and druggable molecular and genetic alterations involved in the pathogenesis of LAC, as well as the obligate drug resistance associated with therapies (Ambrogio *et al*, 2016). This is especially pertinent considering the lack of clinical benefit in advanced NSCLC patients upon targeting downstream signaling pathways with prominent roles in oncogenic KRAS. For instance, systemic blockade of the ERK MAPK pathway with MEK inhibitors trametinib and selumetinib, either as monotherapy or in combination with standard chemotherapy (i.e., docetaxel), has demonstrated limited clinical efficacy. This was associated with acquired resistance-based sustained ERK MAPK signaling, as well as an unfavorable toxicity profile (Kohler *et al*, 2018). Notably, our data here suggest that the specific inhibition of ADAM17 is unlikely to cause such toxicities that are observed upon systemic impairment of ERK MAPK signaling, since reduced ERK1/2 MAPK activation upon global targeting of ADAM17 (e.g., using *Adam17*$^{ex/ex}$ mice) is restricted to the lung (and not other organs, such as liver and muscle). With respect to other pathways associated with oncogenic KRAS activity in the lung, the therapeutic targeting of PI3K has been plagued by dose-limiting toxicities that prevent the use of high concentrations of inhibitors needed to penetrate the tumor and execute anti-cancer activity (Castellano *et al*, 2013). To overcome toxicities associated with interfering with MEK/ERK MAPK and PI3K pathways, the genetic targeting of another downstream effector kinase of mutant KRAS, c-RAF, has recently been shown to suppress *Kras*$^{G12V}$-induced LAC in mice, independent of any effect on MAPK and PI3K pathways (Sanclemente *et al*, 2018). While promising, the caveat of such a finding will be the design and clinical development of highly selective and bioactive c-RAF inhibitors to ensure low toxicity profiles.

Another key finding of our study was that ADAM17, which has at least 80 known substrates, demonstrates biased proteolytic processing of IL-6R (thus shedding sIL-6R) to promote KRAS-driven LAC. In the lung, the preference for IL-6R as a substrate of ADAM17 might be related to the abundance and close proximity

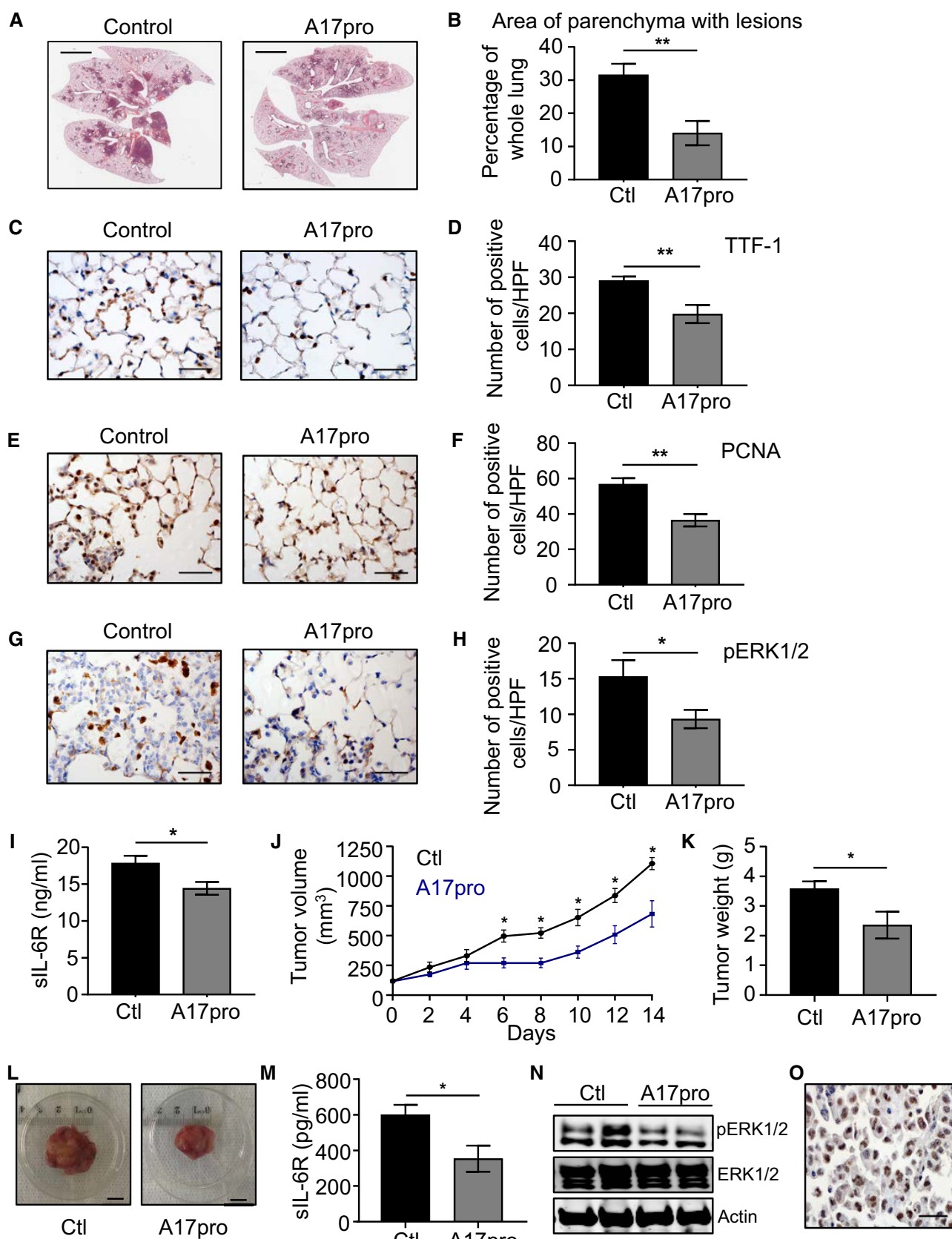

**Figure 7.**

◀

Figure 7.  Anti-cancer activity of the specific ADAM17 prodomain inhibitor, A17pro, in preclinical genetically engineered and PDX KRAS mutant LAC models.

A       Representative images of H&E-stained lung sections from $Kras^{G12D}$ mice treated with either vehicle (control) or the ADAM17 prodomain inhibitor, A17pro (1 mg/kg, 3 times a week), for 6 weeks starting on the day after Cre inhalation. Scale bars, 3 mm.
B       Quantification of lung parenchyma area occupied by tumor lesions in the indicated groups (n = 6 per group) from (A). **P < 0.01, Student's t-test, mean ± SEM.
C–H    (C, E, G) Representative images of lung sections from the indicated mice from (A) stained with antibodies against TTF-1 (C), PCNA (E), and pERK1/2 (G). Scale bars, 100 μm. (D, F, H) Quantification of positive cells/high-power field (HPF) in the lungs of mice from (C), (E), and (G) stained for TTF-1 (D), PCNA (F), and pERK1/2 (H). n = 6 per group. *P < 0.05, **P < 0.01, Student's t-test, mean ± SEM.
I       ELISA of serum sIL-6R protein levels in the indicated mice (n = 6 per group) from (A). *P < 0.05, Student's t-test, mean ± SEM.
J       Tumor volume (mm³) of a KRAS mutant PDX treated with vehicle control (Ctl) or A17pro (1 mg/kg) every second day. Treatment started when tumors reached a volume of 100–150 mm³ after injection of 2 × 10⁶ cells in NSG mice. n = 5 per group. *P < 0.05, Student's t-test, mean ± SEM.
K, L    Weight (grams) (K) and representative images (L) of tumor PDXs from (J). In (K), n = 5 per group. *P < 0.05, Student's t-test. Scale bar, 1 cm, mean ± SEM.
M       ELISA of human sIL-6R protein levels in serum of the indicated mice (n = 5 per group). *P < 0.05, Student's t-test, mean ± SEM.
N       Immunoblots with the indicated antibodies of representative lysates from individual PDX tumors treated with vehicle control or A17pro.
O       Representative image of pADAM17 immunohistochemical staining on an untreated KRAS mutant PDX tumor. Scale bar, 100 μm.

Data information: Exact P values are specified in Appendix Table S4.
Source data are available online for this figure.

of IL-6R to ADAM17, as evidenced by the high level of expression of IL-6R—whose cellular distribution co-localizes with ADAM17—compared to other ADAM17 substrates (e.g., TGFα). We note that the contribution of IL-6, including that via IL-6 trans-signaling which is dependent on sIL-6R, to the oncogenicity of KRAS, has been demonstrated in various KRAS-driven tumors, including LAC (Zhu et al, 2014; Brooks et al, 2016). Although the mechanistic basis by which oncogenic KRAS augments the signaling capacity of IL-6 was not defined in these and other studies, our data here lead us to propose that ADAM17 acts as a molecular bridge between KRAS and IL-6 by upregulating the release of sIL-6R, thus facilitating trans-signaling (in the lung epithelium) via the ERK MAPK pathway. Regarding the latter, since ERK is a key downstream signaling facilitator of oncogenic KRAS-induced cellular proliferation, ADAM17-mediated IL-6 trans-signaling via ERK presents a hitherto unknown pathway specifically utilized by oncogenic KRAS to magnify the signal output of ERK throughout the lung epithelium, thus potentiating a hyper-proliferative state that supports tumorigenesis. It will now be of great interest to further evaluate whether ADAM17-mediated sIL-6R shedding and ERK activation are restricted to mutant KRAS LAC or are also a driver event in other mutant KRAS cancers (e.g., colorectal and pancreatic).

ADAM17-mediated shedding of EGFR family ligands has been linked to oncogenic KRAS-induced pancreatic cancer (Ardito et al, 2012), suggesting that other ADAM17 substrates may also contribute to mutant KRAS-driven LAC. On this note, ADAM17 is associated with shedding of several EGFR family ligands (e.g., EGF, TGFα, and amphiregulin) in NSCLC and other cancers (Zhou et al, 2006; Schmidt et al, 2018). In addition, in human NSCLC cell lines, ADAM17 can upregulate EGFR expression via Notch1, thus modulating cell responsiveness to EGFR ligands (Baumgart et al, 2010). However, blockade of the ADAM17-EGFR axis is likely to have limited efficacy in mutant KRAS-driven LAC, since oncogenic KRAS renders tumors resistant to EGFR inhibitors (Román et al, 2018). In a $Kras^{G12V}$-driven LAC mouse model, indirect evidence that the Notch pathway contributed to LAC was provided by the observed upregulation of an upstream regulator (γ-secretase) and downstream effectors (HES1, DUSP1) of Notch signaling (Maraver et al, 2012). However, contradictory outcomes for the anti-tumor efficacy of genetic or therapeutic targeting of the Notch1 pathway in oncogenic KRAS-driven LAC, along with our own current data showing no alterations to Notch1 activity upon ADAM17 targeting in mutant KRAS LAC models, raise questions over the role of Notch1 as a driver of this oncogenic setting (Ambrogio et al, 2016).

In conclusion, here we identify ADAM17 as a key cooperating factor of oncogenic KRAS in LAC and thus provide a new strategy to indirectly target the oncogenic actions of KRAS in the lung via ADAM17. While our current study revealed the anti-tumor activity of the highly specific ADAM17 inhibitor, A17pro, as a monotherapy in various preclinical mutant KRAS LAC models, one of which was a PDX, there is now the need to evaluate the efficacy of A17pro in combination with existing chemotherapeutics (e.g., cisplatin) (Hirsch et al, 2017). In this respect, despite a potential limitation being the evaluation of A17pro anti-cancer activity in only the one mutant KRAS PDX model, our current findings nonetheless pave the way for future studies to validate the driver role of ADAM17 in additional mutant KRAS LAC PDX models, as well as other PDX models for different lung cancer subtypes (e.g., wild-type KRAS LAC, mutant EGFR LAC, and squamous cell carcinoma). Another consideration is based on the preferential alignment of ADAM17 pro-tumorigenic activity with the processed substrate sIL-6R, whose upregulated levels can be detected in sera from LAC patients, suggesting that sIL-6R could be exploited as a surrogate biomarker for disease-associated ADAM17 activity in the lung (Brooks et al, 2016). Notwithstanding the importance of sIL-6R to mutant KRAS LAC, a comprehensive characterization of the full substrate repertoire of ADAM17 in this disease setting, for instance by quantitative proteomics approaches incorporating terminal amine isotopic labeling of substrates (TAILS), is warranted to refine future biomarker discovery efforts in LAC. Furthermore, we refer to the prospect of targeting ADAM17 versus sIL-6R-dependent trans-signaling in LAC, the latter with existing antibody-mediated therapeutic strategies. The rationale to target trans-signaling in disease, rather than the "classic" IL-6 signaling mode which is dependent on the membrane-bound IL-6R, is based on the opposing roles of IL-6 in either maintaining homeostatic processes (e.g., regulation of B-cell function, the acute phase response, and hematopoiesis), or conversely, driving chronic disease states such as inflammation and cancer (Scheller et al, 2011; Mihara et al, 2012; Jones & Jenkins, 2018). Here, IL-6-mediated homeostatic processes

are facilitated by classic signaling, while the disease-associated functions of IL-6 primarily (or even exclusively) reside with trans-signaling (Hunter & Jones, 2015; Jones & Jenkins, 2018). Notably, the experimental mouse IL-6R antibodies (1F7 and 25F10) used in our current study block IL-6 trans-signaling in the mouse, and have previously been reported to ameliorate tumorigenesis in the $Kras^{G12D}$ LAC model, albeit not as effectively as the robust anti-tumor activity observed here with A17pro (Brooks *et al*, 2016). In addition, existing anti-IL-6R antibody therapies used in the clinic (and for that matter, also those against IL-6), such as tocilizumab and sarilumab, block both protective (i.e., classic) and pathological (i.e., trans) signaling activities of IL-6, causing side effects such as compromised host defense against bacteria (i.e., infections), imbalanced metabolism leading to higher blood cholesterol and triglyceride levels, and increased risk of gastrointestinal tract perforations (Rose-John *et al*, 2017; Jones & Jenkins, 2018). Therefore, highly selective and potent ADAM17 inhibitors such as A17pro used in our current study, which specifically block pathological mouse/human trans-signaling, promise to be more effective in suppressing disease states, including LAC, associated with IL-6 trans-signaling, and with less adverse effects.

# Materials and Methods

### Human biopsies

Resected lung tissues were collected from LAC patients and control LAC-free individuals (Appendix Table S1) from either Monash Medical Centre or the Victorian Cancer Biobank. Tissue samples were snap-frozen in liquid nitrogen or fixed with 4% paraformaldehyde and embedded in paraffin, prior to molecular and histological analyses, respectively. Formal written informed patient consent was obtained prior to blood and tissue collection from all subjects. Studies were conducted in alignment with the ethical principles for medical research involving human subjects set out in the World Medical Association Declaration of Helsinki and Department of Health and Human Services Belmont Report and were approved by the Monash Health Human Research Ethics Committee and the RMIT University Human Ethics Committee.

### Animal models and treatments

$Kras^{LSL-G12D/+}$ mice (DuPage *et al*, 2009) were mated with $Adam17^{ex/ex}$ mice, which are homozygous for a hypomorphic $Adam17$ allele resulting in a dramatic reduction in ADAM17 protein expression (Chalaris *et al*, 2010). The tumor phenotype of the $Kras^{G12D}$ LAC mouse model (maintained on a C57BL/6 × 129Sv background) is induced following intranasal inhalation of $5 \times 10^6$ plaque-forming units of Adenovirus Cre recombinase (Ad-Cre; University of Iowa) by 6-week-old male and female $Kras^{LSL-G12D/+}$ mice (referred to as $Kras^{G12D}$ mice), leading to lung (epithelial)-specific activation of the oncogenic $Kras^{G12D}$ allele (DuPage *et al*, 2009; Brooks *et al*, 2016). Mice were housed under specific pathogen-free conditions and were culled 6 and 12 weeks following inhalation.

The anti-IL6R monoclonal antibodies (mAbs) 25F10 and 1F7 (Lacroix *et al*, 2015), along with IgG control, were intraperitoneal

(i.p.) injected into 6-week-old male and female $Kras^{G12D}$ mice at 10 mg/kg twice weekly for 6 weeks. The ADAM17 prodomain inhibitor (A17pro; 1 mg/kg) was administered by i.p. injection into 6-week-old male and female $Kras^{G12D}$ mice three times a week over 6 weeks. Treatments with mAbs and A17pro started from the day after Ad-Cre inhalation. A single dose of either the p38 MAPK inhibitor (SB203580; 10 mg/kg, Sigma), the ERK1/2 inhibitor (U0126; 10 mg/kg, Cell Signaling Technology), or dimethyl sulfoxide (DMSO) vehicle (control) was i.p. injected into 12-week-old male and female $Kras^{G12D}$ mice at 12 h prior to the 6 weeks of endpoint.

Male and female $Kras^{G12D}$ mice aged 6–8 weeks were lethally irradiated (single 9.5 Gy dose) and reconstituted with $5 \times 10^6$ unfractionated donor bone marrow cells from male and female $Kras^{G12D}$ mice or $Kras^{G12D}:Adam17^{ex/ex}$ mice. Recipient mice were inhaled with Ad-Cre 8 weeks thereafter and were culled 6 weeks following inhalation.

For cell line xenografts, control A549 and ADAM17-deficient A549 cells ($1 \times 10^6$) in Matrigel:phosphate-buffered saline (PBS) (1:1 v/v) were subcutaneously injected into one flank of 10 (*n* = 5 mice per group) 6-week-old male and female NOD.Cg-Prkdc^scid Il2rg^tm1Wjl/SzJ/Arc (NSG) mice (Animal Resources Centre, Canning Vale, Australia). Tumor size was assessed twice weekly using digital calipers, and tumor volumes were calculated with the formula (width $(mm)^2 \times$ length (mm))/2. At the completion of studies, tumors were collected, weighed, and then fixed in 10% formalin.

The generation of a *KRAS* mutant human LAC patient-derived xenograft (PDX) has been described previously (Marini *et al*, 2018). Briefly, a tumor biopsy (poorly differentiated adenocarcinoma, stage T2aN2MX) was collected upon informed consent from a 67-year-old Caucasian man (51 pack year former smoker) for research purposes at Stanford University and was approved by the Stanford Institutional Research Board. Upon screening a series of PDXs for high levels of phosphorylated ADAM17, single-cell suspensions ($2 \times 10^6$) in Matrigel:PBS (1:1 v/v) from a passage-3 PDX were subcutaneously injected into one flank of 12 male and female 6-week-old NSG mice (*n* = 6 mice per group). Once tumors had reached a volume of 100–150 mm$^3$, PBS vehicle or the A17pro inhibitor (1 mg/kg) was administered by i.p. injection every second day, at which time tumor volumes were also determined, over a 2-week period. This study endpoint was determined when tumors in control vehicle-treated mice reached a volume of 1,000 mm$^3$.

All animal experiments were approved by the Monash University Monash Medical Centre "A" and "B" Animal Ethics Committees. Where appropriate, mice were randomly assigned to experimental groups while ensuring equivalent numbers of age-matched males and females. Following experimentation, no animals were excluded from analysis, and no blinding procedure was undertaken. The reporting of mouse studies in this manuscript conforms with the Animal Research: Reporting of *In Vivo* Experiments (ARRIVE) guidelines (Kilkenny *et al*, 2010).

### Cell lines, and cell viability, spheroid aggregate formation, and proliferation assays

Human LAC cell lines A549, NCI-H23, and NCI-H2228 were maintained in DMEM (Invitrogen) supplemented with 10% fetal calf

serum (FCS), 1% penicillin–streptomycin, and 1% L-glutamine (Thermo Fisher Scientific). Cell lines were obtained from the ATCC and were characterized/authenticated via short tandem repeat profiling and passaged in our laboratory for under 6 months after receipt. Cells were routinely tested for mycoplasma contamination (MycoAlert™ PLUS Mycoplasma Detection Kit, Lonza) during the time of experiments. Cells were seeded overnight in 24-well plates with DMEM/10% FCS and 1% L-glutamine, followed by serum starvation for 24 h. Cells then were cultured for 24 h in serum-free DMEM supplemented with DMSO (as a control), A17pro (2 μM), SB203580 (10 μM; Sigma), or GW280264X (2 μM; Aobious).

For cell viability and proliferation assays, $5 \times 10^3$ cells were seeded overnight into triplicate wells (96-well plates) with DMEM/ 10% FCS and 1% L-glutamine, followed by serum starvation for 24 h. Cells then were cultured for 2 days in serum-free DMEM with or without treatments. Cell viability was measured by addition of 0.2 mg/ml of 3-(4,5-dimethylthiazol-2-yl)-2,5-diphenyl tetrazolium bromide (MTT; Sigma-Aldrich) reagent, and cells were incubated for a further 4 h prior to solubilization of crystals with DMSO. Absorbance was measured using a FLUOstar Optima Plate Reader (BMG Labtech) at 560 nm. Cell proliferation was assessed using the CellTiter-Glo Assay (Promega), and luminescence was recorded using a FLUOstar Optima Plate Reader (BMG Labtech). For 3-dimensional spheroid aggregate formation assays, $1 \times 10^4$ A549 cells (parental and ADAM17-targeted) were cultured for 3–5 days in ultra-low attachment 24-well plates (Sigma) in DMEM media supplemented with 10% FCS, EGF, FGF2, FGF10, insulin, and B27 (Sigma); parental cells were treated with or without GW280264X (2 μM) or A17pro (2 μM). Cell aggregates were imaged using an inverted microscope, and then, the area of the cell aggregates (at least 20 per group) was measured using Image J software (National Institutes of Health).

## Primary cell isolation and stimulations

At 2 weeks following Ad-Cre inhalations, 8-week-old $Kras^{G12D}$ mice were euthanized, and the trachea from each mouse was surgically exposed to lavage lungs with 1 ml PBS (3 times). To isolate total immune cells, bronchoalveolar lavage fluid (BALF) was obtained, and total immune cells in BALF were pelleted by centrifugation. Cells were then resuspended in serum-free DMEM and treated with PBS or 100 ng/ml of the potent IL-6 trans-signaling agonist Hyper-IL-6 (Peters et al, 1998) for 1 h. Epithelial cells were isolated from lavaged lungs as described previously (Ruwanpura et al, 2016) and then cultured in 24-well plates (pre-coated with collagen (Sigma)) using DMEM supplemented with FCS, insulin (Thermo Fisher), L-glutamine, and penicillin–streptomycin. After 7 days, epithelial cells were serum starved and treated as above for immune cell cultures.

## CRISPR-driven ADAM17 knockout

Self-complementary oligonucleotides (Sigma) comprising single-guided (sg) RNA sequences against exon 3 of human *ADAM17* were used for the genetic targeting of human *ADAM17* in A549 and NCI-H23 cells, as described previously (Yu et al, 2018). Cloning primers are available upon request.

## Histology and immunohistochemistry (IHC)

Formalin-fixed, paraffin-embedded (FFPE) human lung sections were stained with antibodies against pThr735-ADAM17 (Sigma), as well as pThr202/pTyr204-ERK1/2 and pThr180/pTyr182-p38 MAPK (Cell Signaling Technology). FFPE mouse lung sections were subjected to histologic evaluation by staining with hematoxylin and eosin (H&E), as well as IHC with the following antibodies: B220, CD3, CD45, and Ly6G (BD Biosciences); CD31 and TTF-1 (Abcam); PCNA, cleaved Caspase-3, and pThr202/pTyr204-ERK1/2 (Cell Signaling Technology); pThr735-ADAM17 (Sigma); and F4/80 (Bio-Rad). To quantify cellular staining, digital images of photomicrographs (60× high-power fields) were viewed using Image J software. Positive-staining cells were counted manually ($n = 20$ fields). FFPE xenograft sections were subjected to IHC evaluation by staining with pThr735-ADAM17 (Sigma), pThr202/pTyr204-ERK1/2 (Cell Signaling Technology), and cleaved Notch1 (Abcam) antibodies. Antibody dilutions are indicated in Appendix Table S2.

## Immunofluorescence (IF)

FFPE mouse and human lung sections were subjected to IF evaluation by staining with antibodies against total ADAM17 (rabbit raised, from S. Rose-John; mouse raised, Abcam), CD31 (Abcam), pThr735-ADAM17 (Sigma), CD45 (BD Biosciences), CC10, SPC, IL-6R, TGFα, Nrg1 (Santa Cruz Biotechnology), EGF (R&D Systems), and podoplanin (Abcam). Alexa Fluor conjugates (Life Sciences or Invitrogen) were used as secondary antibodies. Nuclear staining was achieved using 4′,6-diamidino-2-phenylindole (DAPI). Negative controls in the absence of primary antibodies were performed to indicate the level of background autofluorescence. For cell lines, cells were cultured overnight in chambers (Ibidi) supplemented with DMEM/10% FCS and 1% L-glutamine, followed by serum starvation for 24 h. Cells were then subjected to fixation and IF analysis using an anti-pThr735-ADAM17 antibody (Sigma). Antibody dilutions are indicated in Appendix Table S2.

## ELISA and immunoblotting

Total protein lysates were prepared from snap-frozen lung tissues or cell lysates and subjected to ELISA and immunoblotting. Human and mouse IL-6R ELISA sets were purchased from R&D Systems. Immunoblotting was performed with the following antibodies: total ADAM17 (from S. Rose-John), pThr735-ADAM17 (Sigma), Myc, pTyr705-STAT3, pSer727-STAT3, total STAT3, pSer473-AKT, total AKT, pThr202/pTyr204-ERK1/2, total ERK1/2, pThr180/pTyr182-p38 MAPK, total p38 MAPK, pTyr1068-EGFR, total EGFR, pSer82-HSP27 (Cell Signaling Technology), cleaved Notch1 (Abcam), IL-6R, Nrg1, pTyr1289-ErbB3, total ErbB3, TGFα (Santa Cruz Biotechnology), and actin (Sigma). Protein bands were visualized using the Odyssey Infrared Imaging System (LI-COR) and quantified using Image J. Antibody dilutions are indicated in Appendix Table S2.

## RNA isolation and gene expression analyses

Total RNA was isolated from snap-frozen mouse lung tissues using TRIzol (Sigma), and quantitative RT–PCR (qPCR) was performed on cDNA with SYBR Green (Life Technologies) using the 7900HT Fast

### The paper explained

**Problem**

Lung adenocarcinoma (LAC) is the most common form of lung cancer, the leading cause of cancer death worldwide, and is associated with a high risk of tumor re-occurrence following indiscriminate treatment modalities (i.e., surgery, chemotherapy, and/or radiation therapy), as well as poor overall 5-year relative survival rates of 15–20%. Activating mutations in the *KRAS* proto-oncogene are the most extensively studied molecular driver events in LAC and are found in one-third of LAC patients. However, the development and clinical implementation of specific KRAS inhibitors have proven technically challenging, and there is now a pressing need to identify druggable cooperating partners of oncogenic KRAS.

**Results**

In a series of genetically engineered and xenograft (cell line and patient-derived) mutant KRAS-driven LAC animal models, the ADAM17 protease was hyper-phosphorylated. Using these preclinical model systems, the specific genetic and therapeutic targeting of ADAM17, the latter with a non-toxic prodomain inhibitor, suppressed tumor burden by reducing cellular proliferation in tumors. The pro-tumorigenic activity of ADAM17 in LAC was dependent upon its threonine phosphorylation by p38 MAPK, downstream of activated (i.e., oncogenic) KRAS. Among the numerous ADAM17 substrates examined, ADAM17 preferentially co-localized with the substrate, IL-6R, during LAC to release soluble IL-6R that drives IL-6 trans-signaling via the ERK1/2 MAPK pathway. In mutant *KRAS* human LAC tumors, augmented phospho-ADAM17 levels were observed primarily in epithelial rather than immune cells—consistent with a requirement for ADAM17 in mutant KRAS-driven LAC that was independent of bone marrow-derived immune cells—and significantly and positively correlated with activated p38 and ERK1/2 MAPK pathways.

**Impact**

Our study identifies ADAM17 as a key cooperating factor and druggable target in oncogenic KRAS-driven LAC and thus provides the rationale to employ ADAM17-based therapeutic strategies for targeting oncogenic KRAS-addicted cancers.

RT–PCR System (Applied Biosystems). Gene expression data acquisition and analyses were performed using the Sequence Detection System Version 2.4 software (Applied Biosystems). Primer sequences are indicated in Appendix Table S3.

Publicly available RNA sequencing data (IlluminaHiSeq_RNASeqV2 Level 3) from 513 LAC patient tumors, as well as 57 paired tumor/non-tumor LAC patient samples, were obtained from The Cancer Genome Atlas (TCGA) Research Network (https://portal.gdc.cancer.gov/). *ADAM17* mRNA expression was assessed using the DESeq2 R-package (https://www.bioconductor.org/packages/release/bioc/html/DESeq2.html).

**Statistical analyses**

Statistical analyses were performed using GraphPad Prism for Windows version 7.0 software, and significance between experimental group data sets was determined by two-tailed Student's *t*-tests (unpaired or paired, as appropriate), assuming normal distribution. A *P* value of < 0.05 was considered statistically significant, as indicated in figure legends along with experimental sample sizes, where relevant. Experimental sample size estimates were based on

power analyses assuming a significance level (alpha) of 0.05 and a power of 80%, as well as on our previous studies (Brooks *et al*, 2016). Data in figures are expressed as the mean ± standard error of the mean (SEM). Exact *P* values for differences between experimental groups in the presented figures are indicated in Appendix Table S4.

**Expanded View** for this article is available online.

### Acknowledgements

We would like to thank Amanda Vannitamby for preparing human lung tissue sections. This work was supported by the National Health and Medical Research Council (NHMRC) of Australia, the United States Department of Defense (Lung Cancer Research Program Idea Development Award), as well as the Operational Infrastructure Support Program by the Victorian Government of Australia. S.R.J is supported by the Deutsche Forschungsgemeinschaft (DFG) (CRC841, project C1; CRC877, project A1) and by the Excellence Cluster 306 "Inflammation at Interfaces." C.G. is supported by the DFG (SFB877, projects A10 and A14). B.J.J. is supported by a Department of Health | National Health and Medical Research Council (NHMRC) Senior Medical Research Fellowship.

### Author contributions

Conception and design: BJJ; Development of methodology: MIS, SR-J, BJJ; Acquisition of data (provided animals, acquired and managed patients, provided facilities, etc.): MIS, SA, LM, LY, MA, VD, KT, TJ, JAS, ZP, MDT, JJB, DNW, JEC, SB, EA, TK, HE, WF, CG, SR, IS, SR-J, BJJ; Analysis and interpretation of data (e.g., statistical analysis, biostatistics, computational analysis): MIS, TK, HE, IS, SR-J, BJJ; Writing, review, and/or revision of the manuscript: MIS, SB, EA, CG, SR-J, BJJ; Administrative, technical, or material support (i.e., reporting or organizing data, constructing databases): DNW, JEC, BJJ; Study supervision: BJJ.

### Conflict of interest

The authors declare that they have no conflict of interest.

### For more information

(i) https://portal.gdc.cancer.gov/
(ii) https://hudson.org.au/research-group/cancer-and-immune-signalling/
(iii) http://www.uni-kiel.de/Biochemie/
(iv) http://www.uni-kiel.de/Biochemie/sfb877/3rd_period/index.html
(v) https://www.ebi.ac.uk/merops/

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
