## [Review Process File · EMBO Molecular Medicine]

ADAM17 selectively activates the IL-6 trans-signaling/ERK MAPK axis in KRAS-addicted lung cancer

Mohamed I. Saad, Sultan Alhayyani, Louise McLeod, Liang Yu, Mohammad Alanazi, Virginie Deswaerte, Ke Tang, Thierry Jarde, Julian A. Smith, Zdenka Prodanovic, Michelle D. Tate, Jesse J. Balic, D. Neil Watkins, Jason E. Cain, Steven Bozinovski, Elizabeth Algar, Tomohiro Kohmoto, Hiromichi Ebi, Walter Ferlin, Christoph Garbers, Saleela Ruwanpura, Irit Sagi, Stefan Rose-John, Brendan J. Jenkins

Review timeline:

Submission date:	23 October 2018
Editorial Decision:	4 December 2018
Author Comments:	12 December 2018
Editor Reply:	17 December 2018
Revision received:	24 January 2019
Editorial Decision:	1 February 2019
Revision received:	3 February 2019
Accepted:	6 February 2019

Editor: Lise Roth

Transaction Report:

1st Editorial Decision

4 December 2018

Thank you for the submission of your manuscript to EMBO Molecular Medicine. We have now heard back from the two referees who were asked to evaluate your manuscript.

As you will see from the reports below, the referees are overall positive and support publication of the article in EMBO Molecular Medicine pending appropriate revisions. Addressing the reviewers' concerns in full will be necessary for further considering the manuscript in our journal. EMBO Molecular Medicine encourages a single round of revision only and therefore, acceptance or rejection of the manuscript will depend on the completeness of your responses included in the next, final version of the manuscript.

Please also contact us as soon as possible if similar work is published elsewhere. If other work is published, we may not be able to extend the revision period beyond three months.

I look forward to receiving your revised manuscript.

***** Reviewer's comments *****

Referee #1 (Comments on Novelty/Model System for Author):

Experiments have been performed in genetically engineered KrasG12D-driven Lung adenocarcinoma (LAC) models defective or not for ADAM17. Functional results have been confirmed in human LAC cell-lines plus one human patient-derived xenograft model in nude mice. However, I think it would be important to confirm this data by using an additional PDX model. Furthermore, the analyses on the proposed mechanism is not complete and does not entirely justify the conclusion made by the authors. For instance, the authors do not consider the upregulation of ADAM17, which has been reported in human LAC. Biostatistics, according to the methods section, are based on a sufficient number of independent experiments, however some controls in mouse experiments are lacking. This point is specified in the response to the authors.

It has been shown by the authors that IL-6 trans-signaling is an essential downstream event of KRAS-driven LAC and they now add ADAM17 as an important component of this pro-tumoral pathway. They also present additional findings indicating that the targeting of this enzyme could have potential medical impact in LAC. While potentially interesting, it would be important to compare or discuss on the proposed strategy with the one implicating IL-6 antibodies.

Referee #1 (Remarks for Author):

Targeting KRAS-driven lung cancer is a medical problem due to the high difficulties to target KRAS activity. Therefore much effort has been made to develop an effective therapeutic strategy targeting this oncogenic pathway. For instance, several groups including the authors have revealed an important role of IL6 signaling in this tumorigenic process, which could be targeted by a specific immunotherapy. Here the authors reveal that ADAM17 is a central component of this oncogenic pathway and could serve as an attractive target in these tumors. Firstly, they provide evidence that ADAM17 deficiency strongly diminished the tumor burden found in genetically engineered mouse models for KRAS-driven LAC. They next show that this defect was associated with a diminution of tumor cell proliferation. Surprisingly, this pro-tumoral function of ADAM17 involved p38 MAPK-dependent catalytic activation. While ADAM17 signals through the shedding of numerous transmembrane proteins, the authors provide evidence that IL6-R is the main ADAM17 target in this tumorigenic process. This response induces an IL-6 trans-signaling via the ERK1/2 MAPK pathway by an unclear cellular mechanism. Experimental evidence of their model was next provided in human LAC cell-lines as well as in one PDX model. Also, a significant positive correlation between augmented phospho ADAM17 levels and activation of the ERK MAPK pathway was found in a cohort in KRAS LAC samples. Finally, the clinical interest of targeting ADAM17 is these tumors was shown by the anti-cancer effect of the prodomain inhibitor of ADAM17 in both genetic mouse and a PDX LAC model. As such the authors propose that targeting ADAM17 may be of therapeutic interest in KRAS LAC.

In sum, the research described here was overall well-conceived, executed and presented. Yet I feel that additional mechanistic evidence should be incorporated before the manuscript is ready for publication. Also the biological models incorporated in this study should be presented in more detail to improve the clarity and the credibility of the research.

Main points

1. Additional mechanistic evidence for the proposed model should be provided. The authors proposed that an ADAM17-IL6R pathway promotes tumor cell proliferation possibly by acting on the CSC compartment. It would thus be important to confirm this data in vitro by for example using sphere assays.
2. The proposed IL6 trans-signaling is not well defined and the ms does not clearly discuss how the authors envision the proposed mechanism. For instance, it is not known what are the recipient cells responding to this IL6R trans-signaling pathway. In vitro experimental evidence should be provided to support the proposed hypothesis.
3. The authors do not consider the ADAM17 upregulation found in patients with LAC. This point is important as their mouse model do not reflect this potential important process, questioning about its

relevance to the human pathology. It would be, for instance, rewarding to address whether ADAM17 overexpression increases oncogenic KRAS activity by the proposed model and whether the targeting of ADAM17 shows an augmented anti-cancer activity.

4. The proposed mechanism for ADAM17 pro-tumoral activation by oncogenic KRAS is not fully demonstrated. For instance, it is clear what is the signal that triggers p38 MAPK-dependent ADAM17 activation. Besides, ADAM17 targets numerous transmembrane proteins, however the authors tested only few of them. They cannot exclude the involvement of additional targets in ADAM17 pro-tumoral function. This point is important as the authors put strong emphasis on the selective action of ADAM17 on IL6 in these tumors (ie title of the ms). The fact that ADAM17 co-localizes with IL6R in contrast to TGF alpha (which is poorly expressed) is not conclusive.

5. The PDX model incorporated in this study should be better defined and the proposed ADAM17 function in LAC should be confirmed with at least on additional PDX model.

6. While the effect of the combination of ADAM17 prodomain inhibitor with existing therapies is beyond the scope of the study, the authors should at least compare or discuss its anti-cancer activity with the one induced by IL6 antibodies.

7. What about the anti-cancer activity of ADAM17 pro-domain inhibitor in WT KRAS LAC? This point should be tested or at least discussed to see whether this strategy may be restricted to oncogenic KRAS LAC.

Specific points

1. Please better describe in the main text the ADAM17 KO mouse used in this study.
2. Fig 2 please explain the choice of shown targeted genes. While Myc is an obvious cell-cycle candidate, it is not clear why the authors measured Cdc42 expression.
3. Controls (levels in the absence of oncogenic KRAS) are missing in Fig 2, 4 and 5. This point is important to evaluate the impact of ADAM17 on the studied molecular responses.
4. WB in Fig2D is of poor quality.
5. The authors should comment why in the G12D situation the level of pMAPK is moderately (if any) enhanced in comparison to the control situation (see for example Fig4E).
6. Fig 6C is of moderate quality. A quantification of the signals should be incorporated in the ms.
7. Fig 5D is not convincing. Please use an additional ADAM17 target that is well expressed in the studied mouse model, as a negative control.
8. Fig 6D. Please address whether, upon the proposed model, pADAM17 level correlates with p38 activity in LAC patients.
9. Fig 7 Please explain why A17pro demonstrates a moderate inhibitory effect (panel J) in comparison to ADAM17 KO (Fig 6 J).
10. It would be rewarding to address the impact of ADAM17 inhibition on tumor cell growth, survival and angiogenesis in experimental human LAC and discuss these results with to data obtained in the LAC mouse model.

Referee #2 (Comments on Novelty/Model System for Author):

This study by Saad et al., define a new molecular pathway that may drive the pathogenesis of lung cancer. They show that ADAM17 selectively activates the IL-6 trans-signalling/ERK MAPK axis in KRAS-driven lung cancer. KRAS mutations are known to drive tumorigenesis of lung adenocarcinomas (LACs). However, it is poorly targeted and the mechanisms of pathogenesis are incompletely understood. Saad et al., use various models of Kras-driven LAC to show that the protease ADAM17 is required for Kras-induced carcinogenesis.

KRASG12D ADAM17 depleted mice were protected against KRAS -induced tumours, which was gene dose dependent (Fig 1). This was associated with reduced proliferation of tumour cells and reduced inflammation (CD45+ cells, Fig 2). ADAM17 was expressed predominantly in epithelial cell types (Fig 3). The development of tumours was associated with ADAM17 thr735 phosphorylation by p38 MAPK and p38MAP inhibitor reduces pADAM17 (Fig 4). siL-6 is reduced in absence of ADAM17 and anti-IL6R antibody treatment reduced pERK1/2-positive cells (Fig 5). Human KRAS mutant LACs have enhanced activation of the members of the ADAM17-sIL-6R-

ERK1/2 MAPK axis (Fig 6). The ADAM17 prodomain inhibitor A17pro suppressed tumourigenesis in LAC models (Fig 7). These data along with a plethora of supplementary data strongly support the conclusions that:

Inhibition of ADAM17 suppressed tumour burden through reducing cell proliferation
 Tumourigenesis was dependent on 1. ADAM17 threonine phosphorylation by p38 MAPK, and 2. Release of the substrate of ADAM17 soluble IL-6R.

Soluble IL-6R drives IL-6 trans-signaling via the ERK1/2 MAP2 pathway

The requirement for ADAM17 did not depend on bone-marrow-derived haematopoietic immune cells.

In human KRAS mutant LAC phosphor ADAM17 levels and the activation of the ERK MAPK pathway primarily in epithelial cells.

Thus, these studies identify ADAM17 as a druggable target in KRAS-driven LAC.

This is a well-designed and powerful study. It is thoroughly performed and well written. It is novel and provides new insights into the development of LAC.

Referee #2 (Remarks for Author):

This will be of substantial interest to the readership of EMBO Mol Med.
 The authors are experts in this field and the study is very well performed.

Author Comments

12 December 2018

Thank you for the email notification regarding the positive review of our manuscript. We are very pleased that the Referees have appreciated our comprehensive and robust study, and welcome the opportunity to respond to the comments.

On this note, I would like to refer to the following comments by Referee #1 regarding the use of another PDX model, as follows:

"Functional results have been confirmed in human LAC cell-lines plus one human patient-derived xenograft model in nude mice. However, I think it would be important to confirm this data by using an additional PDX model"

"5. The PDX model incorporated in this study should be better defined and the proposed ADAM17 function in LAC should be confirmed with at least on additional PDX model"

In response, we strongly believe against incorporating an additional PDX, as follows:
 - firstly, we argue that our current study using 3 independent in vivo mutant Kras lung adenocarcinoma (LAC) mouse models - namely i) the well characterised genetic KrasG12D strain ("gold standard" mouse model for mutant Kras studies), ii) the A549 human LAC cell line mutant KRAS xenograft model, and iii) a published (in Sci Transl Med by co-authors Watkins and Cain) KRAS mutant LAC PDX - already provides robust and compelling data that the genetic or therapeutic blockade of ADAM17 suppresses mutant KRAS LAC. This is recognised by Referee #2.

- secondly, the practicalities of generating data from another KRAS mutant PDX treated with the A17 prodomain inhibitor within 3 (or even 6) months are most problematic. For instance, this would involve submission to our ethics committee for approval (with the upcoming Xmas/NY shutdown period this would not happen until mid February at the earliest), the thawing and subsequent passaging and expansion of such a PDX in NSG mice (not nude mice as the Referee stated) for experimental cohorts, and then the treatment of such cohorts and subsequent data analysis.

Based on our responses above, I would greatly appreciate your thoughts.

Editor Reply

17 December 2018

I have now received an answer from reviewer #1. While he/she understands the technical concerns that you raised and does not want to delay the publication of your work, he/she remains convinced (as we do) that PDX models are more appropriate to model human pathologies than additional mouse models, and states: "PDX models closer reflect the human pathological situation as compared to additional models incorporated in their ms. For instance, the studied mouse model does not mimic the increase in ADAM17 expression level found in human patients with LAC. Therefore and ideally, results in 2 distinct PDX models would certainly give strong support to their conclusions, specifically for a journal dedicated to molecular medicine such as EMM. I would therefore suggest the authors to discuss such limitation in their discussion and try to be a little more cautious on their conclusion".

1st Revision - authors' response

24 January 2019

(See next page)

We thank the Referees for their insightful and positive comments and suggestions. As detailed below in our point-by-point responses that address all points raised, we believe that we have further strengthened our manuscript. We note that throughout, we have been most mindful of the manuscript character count and thus the need to keep additional text in the manuscript as succinct as possible.

Referee #1 (Comments on Novelty/Model System for Author):

I think it would be important to confirm this data by using an additional PDX model.

We appreciate the basis for the Referee's comment (also the basis for Main point #5 below), whom we thank for further clarifying to us indirectly via email correspondence with the Senior Editor. Firstly, we strongly believe that our current study using 3 independent mutant *Kras* lung adenocarcinoma (LAC) mouse models - namely i) the well characterised genetic *Kras*^{G12D} strain ("gold standard" mouse model for mutant *Kras* studies), ii) the A549 human LAC cell line mutant *KRAS* xenograft model, and iii) a published (in *Sci Transl Med* by co-authors Watkins and Cain) mutant *KRAS* LAC PDX - already provides robust and compelling preclinical data that the genetic or therapeutic blockade of ADAM17 suppresses mutant *KRAS* LAC. This sentiment is recognised by Referee #2. Furthermore, we demonstrate that increased phosphorylation of ADAM17, but not increased expression of ADAM17 (see new Figure EV4B – Expanded View/Supplementary Data file, and our responses to the next Referee point below and to Main point #3), is a feature of both mutant *KRAS* LAC primary patient tumours and the 3 independent published mutant *KRAS* LAC models we employ in our study. Therefore, this provides strong clinical relevance and validation for these mouse models.

Secondly, as per our previous email correspondence with the Senior Editor (and therefore indirectly with the Referee), we refer to the problematic nature of generating data from another

mutant *KRAS* LAC PDX model treated with the ADAM17 prodomain inhibitor within a timely manner (e.g. 3-6 months). Indeed, this would require submission to our ethics committee for approval (with the Xmas/New Year shutdown period this would not happen until mid-February at the earliest), the thawing and subsequent passaging and expansion of such a PDX in NSG mice (not nude mice as the Referee stated) for experimental cohorts, and then the treatment of such cohorts and subsequent data analysis.

Therefore, as per the following suggestion by the Referee, “I would therefore suggest the authors to discuss such limitation in their discussion and try to be a little more cautious on their conclusion”, in the Discussion section on page 20 of the revised manuscript we have now included a statement referring to the potential limitation of employing one PDX and the subsequent need for future studies to validate our findings in additional mutant *KRAS* LAC PDX models, as well as other PDX models for different lung cancer subtypes (e.g. wild-type *KRAS* LAC, mutant EGFR LAC, squamous cell carcinoma).

The authors do not consider the upregulation of ADAM17, which has been reported in human LAC.

Please also see our response below to Main point #3. We are not aware of any published study demonstrating significantly increased expression of ADAM17 in mutant *KRAS* LAC patient tumours, and the Referee has not referred to any such citation in their above comment. Indeed, the only study we are aware of where ADAM17 expression has been examined in lung cancer was by Ni *et al* (*Tumor Biology* 2013;34:1813-1818) - which we already cite - and yet this short report was in a small cohort of NSCLC patients with no stratification for LAC nor *KRAS* mutation status. Notably, new Figure EV4B in the Expanded View/Supplementary Data file now provides compelling clinical data from The Cancer Genome Atlas, comprising n = 513 LAC tumours (n = 438 wild-type *KRAS* and 75 mutant *KRAS*, and n = 57 paired non-

tumour/tumour LAC cases), which demonstrates that *ADAM17* is not over-expressed in either 1) tumour versus non-tumour tissue in LAC, nor 2) *KRAS* mutant versus wild-type LAC tumours. These new data therefore support our original clinical data (immunoblots) in revised Figure EV4C. Furthermore, the increased phosphorylation status of *ADAM17* we discovered in mutant *KRAS* LAC patient tumours (Figure 6) is fully supported by similar findings in the 3 independent and published *KRAS* mutant LAC models we employ (Figures 4, 6 and 7), thus further validating the clinical relevance of our novel mechanistic finding in these 3 models.

Biostatistics, according to the methods section, are based on a sufficient number of independent experiments, however some controls in mouse experiments are lacking. This point is specified in the response to the authors.

We thank the Referee for raising this point, which we now fully address in our response below to Specific point #3 below.

It has been shown by the authors that IL-6 trans-signaling is an essential downstream event of KRAS-driven LAC and they now add ADAM17 as an important component of this pro-tumoral pathway. They also present additional findings indicating that the targeting of this enzyme could have potential medical impact in LAC. While potentially interesting, it would be important to compare or discuss on the proposed strategy with the one implicating IL-6 antibodies.

We thank the Referee for raising this point, which is also directly related to Main point #6 below. It is known that IL-6 has opposing roles in either maintaining homeostatic processes (e.g. regulation of B cell function, the acute phase response, and hematopoiesis), or conversely, driving chronic disease states such as inflammation and cancer (Scheller J *et al. Biochim. Biophys Acta* 2011;1813; 878-888; Mihara M *et al. Clin Sci (Lond)* 2012;122,143-159; Jones

SA, Jenkins BJ. *Nat Immunol Rev* 2018;18:773-789). A wealth of literature also reveals that the homeostatic processes governed by IL-6 depend on its signalling via the membrane-bound IL-6 receptor (mIL-6R), which is referred to as classic signalling (Hunter CA, Jones SA. *Nat Immunol* 2015;16:448-457). By contrast, the disease-associated functions of IL-6 primarily (or even exclusively) reside with trans-signalling via soluble (s) IL-6R, which is produced by ADAM17-mediated proteolytic cleavage of mIL-6R (Hunter CA, Jones SA. *Nat Immunol* 2015;16:448-457; Riethmueller S *et al. PLoS Biol* 2017;15:e2000080; Jones SA, Jenkins BJ. *Nat Immunol Rev* 2018;18:773-789). Notably, the experimental mouse IL-6R antibodies (1F7 and 25F10) used in our current study block IL-6 trans-signalling in the mouse, and have previously been reported (cited reference, Brooks *et al*, 2016) to ameliorate tumorigenesis in the *Kras*^{G12D} LAC model, albeit not as effectively as the robust anti-tumor activity observed here with A17pro. In addition, existing anti-IL-6R antibody therapies used in the clinic (and for that matter, also those against IL-6), such as tocilizumab and sarilumab, block both protective (i.e. classic) and pathological (i.e. trans) signalling activities of IL-6, causing side effects such as compromised host defence against bacteria (i.e. infections), imbalanced metabolism leading to higher blood cholesterol and triglyceride levels, and increased risk of gastrointestinal tract perforations (Rose-John S *et al. Nat Rev Rheumatol* 2017;13:399-409; Jones SA, Jenkins BJ. *Nat Immunol Rev* 2018;18:773-789). Therefore, highly selective and potent ADAM17 inhibitors such as the A17pro prodomain inhibitor used in our current study, which specifically block pathological trans-signalling (in mouse and human), promise to be more effective in suppressing disease states, including LAC, associated with IL-6 trans-signalling, and with less adverse effects.

As requested by the Referee, we now include the above text discussing this matter, along with new cited references, in the Discussion section on pages 20 and 21 of the revised manuscript.

Referee #1 (Remarks for Author):

Main points

1. Additional mechanistic evidence for the proposed model should be provided. The authors proposed that an ADAM17-IL6R pathway promotes tumor cell proliferation possibly by acting on the CSC compartment. It would thus be important to confirm this data *in vitro* by for example using sphere assays.

We thank the Referee for this astute comment. As requested by the Referee, we have now performed *in vitro* spheroid aggregate formation assays on A549 mutant *KRAS* human LAC cells, as presented in new Figure EV5G and 5H (see below). These data show that the formation of 3D spheroid aggregates was dramatically diminished in A549 cells either deficient in ADAM17 or treated with the ADAM17 inhibitors A17pro or GW280264X (dual ADAM10/17 inhibitor), thus suggesting that ADAM17 can also impact on the proliferative capacity of cancer stem (i.e. initiating) cells in mutant *KRAS* LAC. These new data are stated in the text of the revised manuscript on page 14 of the Results, and methodology for these spheroid aggregate formation assays is provided on page 25 in the Materials and Methods section.

2. The proposed IL6 trans-signaling is not well defined and the ms does not clearly discuss how the authors envision the proposed mechanism. For instance, it is not known what are the recipient cells responding to this IL6R trans-signaling pathway. In vitro experimental evidence should be provided to support the proposed hypothesis.

In Figures 5D and 5E of the revised manuscript we demonstrate that antibody-mediated blockade of sIL-6R-driven IL-6 trans-signalling (which suppresses the LAC phenotype; Brooks *et al*, 2016) impairs the ERK MAPK pathway in the lungs of *Kras*^{G12D} mice. Furthermore, in Figure 4L of the revised manuscript we demonstrate that only the ERK MAPK pathway is suppressed upon genetic targeting of ADAM17 in *Kras*^{G12D}:*Adam17*^{ex/ex} mice (in which IL-6 trans-signalling is suppressed). Together, these observations define the ERK MAPK pathway as the key downstream signalling cascade of the ADAM17/IL-6 trans-signalling axis in oncogenic Kras-induced LAC. Indeed, on page 19 in the Discussion section of the revised manuscript we clearly discuss the proposed mechanism with the following text: “our data here lead us to propose that ADAM17 acts as a molecular bridge between KRAS and IL-6 by upregulating the release of sIL-6R, thus facilitating trans-signaling (in the lung epithelium) via the ERK MAPK pathway. Regarding the latter, since ERK is a key downstream signaling facilitator of oncogenic KRAS-induced cellular proliferation, ADAM17-mediated IL-6 trans-signaling via ERK presents a hitherto unknown pathway specifically utilised by oncogenic KRAS to magnify the signal output of ERK throughout the lung epithelium, thus potentiating a hyper-proliferative state that supports tumorigenesis”.

With respect to the Referee’s comment about providing *in vitro* experimental evidence regarding the recipient cells responding to IL-6R trans-signalling, as requested, we now present these data in the new Figure 5F of the revised manuscript (see also below). Specifically, we show that the potent IL-6 trans-signalling agonist Hyper-IL-6 (new reference Peters *et al*, 1998) upregulated ERK1/2 MAPK signalling in epithelial cells, but not immune cells, isolated from

mouse lungs harbouring the activated *Kras*^{G12D} allele. In the revised manuscript, these new data are referred to in the Results section on page 12, and the methodology associated with this experiment is included in the new paragraph “*Primary cell isolation and stimulations*” on page 25 of the Materials and Methods section.

F Representative immunoblots with the indicated antibodies on whole cell lysates from primary immune and epithelial cells isolated from the lungs of mice (n = 3) harbouring the oncogenic *Kras*^{G12D} allele that were stimulated with either PBS (-) or 100ng/ml Hyper-IL-6 (+). Graphs depict semi-quantitative densitometry of pERK1/2 MAPK protein levels (relative to total ERK1/2 MAPK) in the stimulated immune and epithelial cells (n = 2 experiments).

3. The authors do not consider the ADAM17 upregulation found in patients with LAC. This point is important as their mouse model do not reflect this potential important process, questioning about its relevance to the human pathology. It would be, for instance, rewarding to address whether ADAM17 overexpression increases oncogenic KRAS activity by the proposed model and whether the targeting of ADAM17 shows an augmented anti-cancer activity.

Please also see our response above to the Referee’s second comment in “Comments on Novelty/Model System for Author”. As stated above, we are not aware of any published study demonstrating significantly increased expression of ADAM17 in mutant *KRAS* LAC patient tumours. Rather, as we cite in our manuscript (Ni *et al Tumor Biology* 2013:34;1813-1818), ADAM17 expression has only been examined in a small cohort of NSCLC patients, yet with

no stratification for LAC nor *KRAS* mutation status. Notably, in the new Figure EV4B (see also below) of our revised manuscript, we now provide compelling clinical data from The Cancer Genome Atlas (TCGA) Research Network comprising 513 LAC patient tumours (n = 438 wild-type *KRAS* and 75 mutant *KRAS*, and n = 57 paired non-tumour/tumour LAC cases) demonstrating that *ADAM17* is not over-expressed in either 1) tumour versus non-tumour tissue in LAC, nor 2) *KRAS* mutant versus wild-type LAC tumours. These new data also support our original clinical data (immunoblots) in revised Figure EV4C. Importantly, we also note that the increased phosphorylation status of ADAM17 we have discovered in mutant *KRAS* LAC patient tumours (Figure 6) is fully supported by similar findings in the 3 independent and published *KRAS* mutant LAC models we employ (Figures 4, 6 and 7), thus further validating the clinical relevance of our novel mechanistic finding in these 3 models to human LAC pathology.

With respect to the comment “rewarding to address whether ADAM17 overexpression increases oncogenic *KRAS* activity by the proposed model”, firstly, as stated above, we demonstrate that ADAM17 is not over-expressed in mutant *KRAS* human LAC tumours. Secondly, it has already been published that transgenic over-expression of ADAM17 in mice, including in the lung, does not result in increased shedding (i.e. protease) activity, thus indicating that ADAM17 activity is not dependent on its transcriptional regulation (Yoda M *et al. PLoS One* 2013;8:e54412). Therefore, it is unclear what value such an over-expression experiment would provide. With respect to the additional comment by the Referee, “whether the targeting of ADAM17 shows an augmented anti-cancer activity”, we already demonstrate that the targeting of ADAM17 in 3 independent mutant *KRAS* LAC models displaying elevated pADAM17 has robust anti-cancer activity (Figures 1, 4, 6 and 7).

In the revised manuscript, we now refer to our new ADAM17 expression data from the LAC patient TCGA database (new Figure EV4B; see below) in the Results section on page 13, as well as the associated methodology to generate these data in the Materials and Methods section on page 27.

4. The proposed mechanism for ADAM17 pro-tumoral activation by oncogenic KRAS is not fully demonstrated. For instance, it is clear what is the signal that triggers p38 MAPK-dependent ADAM17 activation. Besides, ADAM17 targets numerous transmembrane proteins, however the authors tested only few of them. They cannot exclude the involvement of additional targets in ADAM17 pro-tumoral function. This point is important as the authors put strong emphasis on the selective action of ADAM17 on IL6 in these tumors (ie title of the ms). The fact that ADAM17 co-localizes with IL6R in contrast to TGF alpha (which is poorly expressed) is not conclusive.

Please also see our new data in new Figure 5C which is provided below in response to the related Specific point #7 by the Referee. We have now performed immunofluorescence staining of *Kras*^{G12D} mouse lung sections for additional ADAM17 substrates implicated in oncogenesis, namely Notch1 and Nrg1, as well as EGF as a control non-ADAM17 substrate. These new data are referred to on page 12 in the Results section of the revised manuscript, and reveal that

neither Notch1 nor Nrg1 co-localise with ADAM17, unlike IL-6R. Therefore, these data further support our finding that ADAM17 preferentially employs IL-6R as a major substrate in oncogenic Kras-induced LAC.

We also note that, as intimated by the Referee, ADAM17 has numerous (over 70) known substrates, and while we provide compelling data that IL-6R is preferentially employed by ADAM17 as a substrate in oncogenic Kras-induced LAC, we agree with the Referee that it remains possible that other substrates could also play a role, albeit minor, in Kras-induced LAC pathogenesis. Indeed, for this very reason, the following original statement is included in the Discussion section on page 20 of the revised manuscript: “*Notwithstanding the importance of sIL-6R to mutant KRAS LAC, a comprehensive characterization of the full substrate repertoire of ADAM17 in this disease setting, for instance by quantitative proteomics approaches incorporating terminal amine isotopic labelling of substrates (TAILS), is warranted to refine future biomarker discovery efforts in LAC.*”

5. The PDX model incorporated in this study should be better defined and the proposed ADAM17 function in LAC should be confirmed with at least on additional PDX model.

Firstly, the generation of the mutant *KRAS* LAC PDX model employed in this study has been published, and is referred to on page 23 in the Materials and Methods section of the revised manuscript (Marini *et al*, 2018). In addition, as requested by the Referee, we have included an additional statement on page 23 in the Materials and Methods section of the revised manuscript providing further information on the primary patient sample used to derive this PDX.

Secondly, please see our detailed response above to the Referee’s first comment in “Comments on Novelty/Model System for Author” which states our rationale and justification for not including an additional PDX model. Rather, as suggested by the Referee via email correspondence with the senior Editor (“I would therefore suggest the authors to discuss such

limitation in their discussion and try to be a little more cautious on their conclusion”), we have now included a statement in the Discussion section on page 20 of the revised manuscript referring to the potential limitation of employing one PDX and the subsequent need for future studies to validate our findings in additional mutant *KRAS* LAC PDX models, as well as other PDX models for different lung cancer subtypes.

6. While the effect of the combination of ADAM17 prodomain inhibitor with existing therapies is beyond the scope of the study, the authors should at least compare or discuss its anti-cancer activity with the one induced by IL6 antibodies.

Please see our detailed response above under “Comments on Novelty/Model System for Author” to the related fourth point raised by the Referee. As requested, we have now included expansive new text comparing the use of antibodies targeting IL-6 signalling versus ADAM17 targeting with the A17pro inhibitor on pages 20 and 21 in the Discussion section of the revised manuscript.

7. What about the anti-cancer activity of ADAM17 pro-domain inhibitor in WT KRAS LAC? This point should be tested or at least discussed to see whether this strategy may be restricted to oncogenic KRAS LAC.

We agree with the Referee that our compelling data demonstrating the anti-cancer activity upon ADAM17 targeting in mutant *KRAS* LAC now paves the way for further studies to investigate whether targeting ADAM17 (e.g. with the A17pro inhibitor) is effective in other subtypes of lung cancer, including wild-type *KRAS* LAC. However, we trust the Referee can appreciate that experimentally evaluating the anti-cancer activity of ADAM17 targeting in additional lung cancer subtypes is well and truly beyond the scope of our current comprehensive study on mutant *KRAS* LAC. For this reason, and as suggested by the Referee, we have included the

following statement on page 20 of the Discussion section of the revised manuscript, which strategically also incorporates our response to Main point #5 above: “*In this respect, despite a potential limitation being the evaluation of A17pro anti-cancer activity in only the one mutant KRAS PDX model, our current findings nonetheless pave the way for future studies to validate the driver role of ADAM17 in additional mutant KRAS LAC PDX models, as well as other PDX models for different lung cancer subtypes (e.g. wild-type KRAS LAC, mutant EGFR LAC, squamous cell carcinoma).*”

Specific points

1. Please better describe in the main text the ADAM17 KO mouse used in this study.

We assume the Referee is referring to the *Adam17^{ex/ex}* mouse strain which was first described in detail in 2010 (Chalaris *et al*, 2010), and has since been widely published in the literature. Nonetheless, to address the Referee’s comment, we have now included an additional statement (first paragraph of the Results section on page 7 of the revised manuscript) describing the already published genetic modification in this mouse strain.

2. Fig 2 please explain the choice of shown targeted genes. While Myc is an obvious cell-cycle candidate, it is not clear why the authors measured Cdc42 expression.

Each of the selected genes assessed in Figure 2C, namely *Ccnd1*, *Ccnb1*, *Cdc42* and *Myc* are i) involved in cell cycle progression, and thus cellular proliferation, and ii) upregulated and implicated in human LAC (see also new references Chen *et al*, 2012; Xu *et al*, 2015; Gao *et al*, 2018). In Figures 2A-C, our data demonstrate that the reduced expression of these genes in the *Kras^{G12D}:Adam17^{ex/ex}* mice bearing smaller lung lesions (compared to parental *Kras^{G12D}* mice) is accompanied by lower cellular proliferation levels. Accordingly, these data further support our discovery that ADAM17 promotes oncogenic Kras-driven LAC by augmenting cellular

proliferation. In the revised manuscript, we have now amended the Results section (page 8) to include this additional information justifying the selection of these genes, as well as including additional references (Chen *et al*, 2012; Xu *et al*, 2015; Gao *et al*, 2018) implicating their upregulated expression in the pathogenesis of human LAC.

We also note that, as per the requested inclusion by the Referee of control *Kras*^{WT} mouse data in the revised Figure 2C (see Specific point #3 immediately below), these data also confirm that each of these cell cycle progression regulatory genes is significantly upregulated in *Kras*^{G12D} mouse lungs.

3. Controls (levels in the absence of oncogenic KRAS) are missing in Fig 2, 4 and 5. This point is important to evaluate the impact of ADAM17 on the studied molecular responses.

As requested by the Referee, where appropriate, we have now included data for control *Kras*^{WT} mice (i.e. absence of oncogenic *Kras*) in the revised Figure 2 (panel C) and Figure 5 (panels A and B) (see also below).

For Figure 4 (panel L), we point out that the control *Kras*^{WT} mouse blots for phosphorylated and total p38 and ERK MAPKs are already presented in panel E. With respect to the *Kras*^{WT} blots for phosphorylated and total STAT3 and AKT, these are provided below for the Referee only. Since the levels of these signalling molecules are unchanged among the various genotypes, and because Figure 4 is already quite dense with a large amount of data, we strongly believe it sufficient to include the following amended text in the Results section of the revised manuscript on page 11: “Immunoblot analysis revealed a striking reduction in ERK1/2 MAPK phosphorylation in lungs of *Kras*^{G12D:Adam17^{ex/ex} versus parental *Kras*^{G12D} mice, whereas the phosphorylation status of other intracellular signaling mediators (STAT3, AKT)}

remained unchanged upon modulating either *Kras* activation or *ADAM17* expression levels in the lung (Fig 4L and data not shown).”

4. WB in Fig2D is of poor quality.

As requested by the Referee, new blots are presented in the revised Figure 2D, along with the updated quantifications in the graph in revised Figure 2E (see also below).

5. The authors should comment why in the G12D situation the level of pMAPK is moderately (if any) enhanced in comparison to the control situation (see for example Fig4E).

It is well established that despite the constitutive activation of mutant *Kras*^{G12D} in numerous mouse tissues (e.g. lung, colon) and primary cell types (e.g. embryonic fibroblasts, bone marrow cells), as well as human cancer (including lung) cell lines, detectable levels of pERK

MAPK are either unaltered or only minimally increased compared to those in tissues and cells harbouring wild-type *Kras* (Tuveson DA *et al*, *Cancer Cell*. 2004;5:375-87; Shi L *et al*, *Cell Death Dis*. 2018;9:219; Cicchini M *et al*, *Cell Rep*. 2017;18:1958-1969; Diaz-Flores E *et al*, *Sci Signal*. 2013;6:ra105). Indeed, lung lesions of the *Kras*^{G12D}-driven LAC model are characterised by low constitutive levels of pERK MAPK that are comparable to levels observed in LAC-free wild-type mouse lungs (Tuveson DA *et al*, *Cancer Cell*. 2004;5:375-87; Cicchini M *et al*, *Cell Rep*. 2017;18:1958-1969). Therefore, our data in Figure 4E showing only “moderately (if any) enhanced” pERK1/2 MAPK levels in the lungs of *Kras*^{G12D} mice are consistent with the literature. Being mindful of the need to keep the manuscript as succinct and concise as possible, we have therefore now included a brief statement citing the above references (i.e. Tuveson DA *et al*, *Cancer Cell*; Cicchini M *et al*, *Cell Rep*) in the Results section on page 10 of the revised manuscript.

For the Referee only, we note that while the mechanistic basis for low pERK MAPK activity downstream of oncogenic *Kras* remains unresolved, since excessive ERK MAPK activity can reduce cell viability, it has been proposed that stringent negative feedback mechanisms in cancer cells may attenuate hyper-active ERK MAPK signalling, thus preventing growth arrest (Ryan MB *et al*, *Trends Cancer*. 2015;1:183-198 ; Diaz-Flores E *et al*, *Sci Signal*. 2013;6:ra105). An alternative notion is that maximal ERK MAPK signalling downstream of activated *Kras* requires that *Kras* both bind to GTP and be localized (dependent on PLC- γ and PI3K) at signalling complexes on ligand-activated cytokine receptors. In this scenario, and in the context of oncogenic *Kras*^{G12D}, *Kras* only accumulates in its GTP-bound state independent of ligand-activated receptors, thus leading to low pERK MAPK levels downstream of oncogenic *Kras*^{G12D} (Diaz-Flores E *et al*, *Sci Signal*. 2013;6:ra105).

6. Fig 6C is of moderate quality. A quantification of the signals should be incorporated in the ms.

We thank the Referee (via email correspondence with the Senior Editor) for the requested clarification on this comment, and we acknowledge the Referee was in fact referring to Figure 6E (not “Fig 6C” as mistakenly stated above in their original comment). Therefore, as suggested by the Referee, we now have performed densitometric quantification of the blots, and the quantified signals are presented in graph format in the new Figure 6G (see also below).

7. Fig 5D is not convincing. Please use an additional ADAM17 target that is well expressed in the studied mouse model, as a negative control.

Please also refer to our response to the related Main point #4 above. As requested by the Referee, we have now performed immunofluorescence staining of *Kras*^{G12D} mouse lung sections for additional ADAM17 substrates implicated in oncogenesis, namely Notch1 and Nrg1, as well as EGF as a control non-ADAM17 substrate. These new data are presented in the new Figure 5C (see also below) and are referred to on page 12 in the Results section of the revised manuscript, and reveal that neither Notch1 nor Nrg1 co-localise with ADAM17, unlike IL-6R. Therefore, these data further support our finding that ADAM17 preferentially employs IL-6R as a major substrate in oncogenic *Kras*-induced LAC.

8. Fig 6D. Please address whether, upon the proposed model, pADAM17 level correlates with p38 activity in LAC patients.

As requested by the Referee, in new Figure 6E (see below) we have now included a graph demonstrating that, in support of our hypothesis, pADAM17 levels do indeed significantly correlate also with pp38 MAPK levels in the lungs of patients. These new data are mentioned in the text on page 13 of the Results section of the revised manuscript.

9. Fig 7 Please explain why A17pro demonstrates a moderate inhibitory effect (panel J) in comparison to ADAM17 KO (Fig 6 J).

The original Figure 6J (now revised Figure 6L) demonstrates the significant ($P < 0.05$; day 6 through to day 21 – end of experiment) anti-tumour effect of CRISPR/Cas9-mediated ADAM17 ablation in the A549 human LAC cell line-derived xenograft. Figure 7J also demonstrates the significant ($P < 0.05$; day 6 through to day 14 – end of experiment) anti-tumour effect of the A17pro inhibitor in a human LAC patient-derived xenograft (PDX). Therefore, key differences in these two experiments are i) the growth kinetics of the cell line-derived xenograft versus PDX, and ii) the genetic (CRISPR/Cas9) versus inhibitor (A17pro) approaches to target ADAM17. As such, it is not possible to directly compare the magnitude of the significant suppression in tumour growth observed upon these two distinct approaches to target ADAM17 in two different xenograft models. Rather, and most importantly, we reveal that A17pro demonstrates a strong and significant tumour inhibitory effect in the PDX (Figure 7J), and for that matter the *Kras*^{G12D} model (Figures 7A and 7B).

10. It would be rewarding to address the impact of ADAM17 inhibition on tumor cell growth, survival and angiogenesis in experimental human LAC and discuss these results with to data obtained in the LAC mouse model.

As requested by the Referee, we have now performed immunohistochemistry with PCNA (proliferation), cleaved Caspase-3 (apoptosis) and CD31 (angiogenesis) in the control versus A17pro treated *KRAS* mutant LAC PDX. As is shown in the new Appendix Figure S3 (see also below), consistent with the effect of A17pro on suppressing tumour cell proliferation in the *Kras*^{G12D} LAC model, A17pro treatment of the *KRAS* mutant LAC PDX significantly suppressed cell proliferation, but had no effect on tumour cell apoptosis nor no angiogenesis.

These new data are briefly discussed with respect to the *Kras*^{G12D} mutant LAC model in the Results section on page 15 of the revised manuscript.

Appendix Figure S3. Reduced proliferation, but not apoptosis or angiogenesis, is associated with suppressed tumorigenesis in an A17pro-treated *KRAS* mutant LAC PDX.

A, C, E Representative high power photomicrographs of lung cross-sections from a *KRAS* mutant PDX treated with vehicle control (Ctl) or A17pro (1 mg/kg) every second day that were stained with antibodies against PCNA (A), cleaved Caspase-3 (C) and CD31 (E). Scale bars, 100 μ m.

B, D, F Quantification of positive cells per high power field (HPF) stained for PCNA (B), cleaved Caspase-3 (D) and CD31 (F) in the above treated *KRAS* mutant PDX. Data are presented as the mean \pm SEM (n = 5 per genotype). **P* < 0.05, Student's *t*-test.

Referee #2 (Comments on Novelty/Model System for Author):

These data along with a plethora of supplementary data strongly support the conclusions..... these studies identify ADAM17 as a druggable target in KRAS-driven LAC. This is a well-designed and powerful study. It is thoroughly performed and well written. It is novel and provides new insights into the development of LAC.

We deeply thank the Referee for their positive comments and appreciation of the novelty of our comprehensive study uncovering the role of the ADAM17-sIL-6R-ERK1/2 MAPK axis in KRAS-driven LAC.

Referee #2 (Remarks for Author):

This will be of substantial interest to the readership of EMBO Mol Med. The authors are experts in this field and the study is very well performed.

We again thank the Referee their positive comments regarding the significance and quality of our comprehensive study.

2nd Editorial Decision

1 February 2019

Thank you for the submission of your revised manuscript to EMBO Molecular Medicine. We have now received the enclosed report from the referee who was asked to re-assess it. As you will see, this referee is now supportive of publication. I am thus pleased to inform you that we will be able to accept your manuscript pending the minor editorial amendments.

***** Reviewer's comments *****

Referee #1 (Remarks for Author):

The quality and the clarity of the manuscript have been very much improved and the additional experiments further support the proposed model. I therefore recommend the publication of this work in EMM.

2nd Revision - authors' response

3 February 2019

Authors made the requested editorial changes.